# Anesthetics fragment hippocampal network activity, alter spine dynamics, and affect memory consolidation

Wei Yang[1,☯], Mattia Chini[2,☯], Jastyn A. Pöpplau[2], Andrey Formozov[1], Alexander Dieter[1], Patrick Piechocinski[1], Cynthia Rais[1], Fabio Morellini[3], Olaf Sporns[4,5], Ileana L. Hanganu-Opatz[2], J. Simon Wiegert[1]*

1 Research Group Synaptic Wiring and Information Processing, Center for Molecular Neurobiology Hamburg, University Medical Center Hamburg-Eppendorf, Hamburg, Germany, 2 Institute of Developmental Neurophysiology, Center for Molecular Neurobiology Hamburg, University Medical Center Hamburg-Eppendorf, Hamburg, Germany, 3 Research Group Behavioral Biology, Center for Molecular Neurobiology Hamburg, University Medical Center Hamburg-Eppendorf, Hamburg, Germany, 4 Department of Psychological and Brain Sciences, Indiana University, Bloomington, Indiana, United States of America, 5 Indiana University Network Science Institute, Indiana University, Bloomington, Indiana, United States of America

☯ These authors contributed equally to this work.
* simon.wiegert@zmnh.uni-hamburg.de

**Data Availability Statement:** The code generated during this study is available at https://github.com/mchini/Yang_Chini_et_al The datasets underlying the figures are available at: https://github.com/

## Abstract

General anesthesia is characterized by reversible loss of consciousness accompanied by transient amnesia. Yet, long-term memory impairment is an undesirable side effect. How different types of general anesthetics (GAs) affect the hippocampus, a brain region central to memory formation and consolidation, is poorly understood. Using extracellular recordings, chronic 2-photon imaging, and behavioral analysis, we monitor the effects of isoflurane (Iso), medetomidine/midazolam/fentanyl (MMF), and ketamine/xylazine (Keta/Xyl) on network activity and structural spine dynamics in the hippocampal CA1 area of adult mice. GAs robustly reduced spiking activity, decorrelated cellular ensembles, albeit with distinct activity signatures, and altered spine dynamics. CA1 network activity under all 3 anesthetics was different to natural sleep. Iso anesthesia most closely resembled unperturbed activity during wakefulness and sleep, and network alterations recovered more readily than with Keta/Xyl and MMF. Correspondingly, memory consolidation was impaired after exposure to Keta/Xyl and MMF, but not Iso. Thus, different anesthetics distinctly alter hippocampal network dynamics, synaptic connectivity, and memory consolidation, with implications for GA strategy appraisal in animal research and clinical settings.

## Introduction

General anesthesia is a drug-induced, reversible behavioral condition encompassing unconsciousness, amnesia, sedation, immobility, and analgesia [1,2]. Together, these aspects represent a state where surgery can be tolerated without the requirement for further drugs [2]. The

mchini/Yang_Chini_et_al/tree/master/Stats_Dataset_(R)/datasets The calcium imaging and electrophysiology data sets generated during this study are available at https://gin.g-node.org/SW_lab/Anesthesia_CA1.

**Funding:** This work was funded by the Deutsche Forschungsgemeinschaft (DFG, SPP1926, FOR2419/P6, SFB936/B8 to J.S.W., SPP 1665/Ha 4466/10-1/Ha4466/12-1, SFB 936/B5 to I.L.H.-O., SFB 936/B7 to F.M.), the European Research Council (ERC2016-StG-714762 to J.S.W., ERC-2015-CoG 681577 to I.L.H.-O.), the German Academic Exchange Service (DAAD, STG/19/5744091 to A.F.), and the Chinese Scholarship Council (CSC 201606210129 to W.Y.). The funders had no role in study design, data collection and analysis, decision to publish, or preparation of the manuscript.

**Competing interests:** The authors have declared that no competing interests exist.

**Abbreviations:** AMPAR, α-amino-3-hydroxy-5-methyl-4-isoxazolepropionic acid receptor; AP, affinity propagation; APC, affinity propagation clustering; dCA1, CA1 area of the dorsal hippocampus; E/I, excitation/inhibition; EMG, electromyography; FAB, flumazenil/atipamezole/buprenorphine; FOV, field of view; GA, general anesthetic; GABAR, γ-aminobutyric acid receptor; ID, identity; Iso, isoflurane; Keta/Xyl, ketamine/xylazine; LFP, local field potential; MMF, medetomidine/midazolam/fentanyl; NMDAR, N-methyl-D-aspartate receptor; NREM, non-rapid eye movement; PAC, phase-amplitude coupling; PCA, principal component analysis; PPC, pairwise phase consistency; REM, rapid eye movement; ROI, region of interest; S.L.M., stratum lacunosum moleculare; S.O., stratum oriens; S.R., stratum radiatum; STTC, spike time tiling coefficient; SUA, single-unit activity; tSNE, t-distributed stochastic neighbor embedding.

behavioral effects of general anesthetics (GAs) are dose dependent. At clinical (i.e., highest) dosage, they should induce unconsciousness, even though experimental evidence of this phenomenon is challenging to collect (in the absence of a verifiable consciousness theory). At lower doses, some GAs cause unresponsiveness and loss of working memory, phenomena that have both been hypothesized to potentially confound the apparent loss of consciousness [3,4]. At much lower doses still, GAs cause profound retrograde amnesia. When general anesthesia fails to induce such behavioral effects, intraoperative awareness ensues, a condition that is associated with long-term adverse health consequences [5]. While loss of memory is required during anesthesia administration, so that no memories of the surgical procedure are formed [1,6], long-term impairment of retrograde or anterograde memories is not desired. Although general anesthesia is generally considered a safe procedure, growing literature points to the possibility of long-term negative effects on the central nervous system [7]. This is particularly true for specific categories of patients, such as the elderly, infants, and children [7]. Among the observed side effects, the most common are postoperative cognitive dysfunction syndromes, including postoperative delirium and postoperative cognitive decline. Postoperative cognitive disturbances are positively correlated with the duration of anesthesia, and a single exposure to GAs can cause retrograde and anterograde memory deficits that persist for days to weeks in rodent models [8]. These aspects point to a generalized action of GAs on the memory system.

Given that amnesia is a fundamental part of general anesthesia and that the hippocampus controls memory formation and consolidation, it is important to understand how anesthetics affect hippocampal function and how this compares to sleep—a naturally occurring state of unconsciousness. Together with the subiculum, the CA1 area constitutes the main hippocampal output region. CA1 pyramidal cells receive excitatory synaptic input mainly from CA3 (in stratum oriens [S.O.] and stratum radiatum [S.R.]) and layer 3 of entorhinal cortex (in stratum lacunosum moleculare [S.L.M.]), relaying information about the internal state of the animal and sensory inputs from the external environment, respectively [9]. Inputs along these pathways are processed in an integrative manner in CA1 [10]. Thus, CA1 pyramidal cells have been suggested to be a site of sensory integration, with synaptic spines as a possible location of memory storage [11–14]. Moreover, dynamic modulation of spine stability has been linked to synaptic plasticity [15–18]. Synaptic plasticity, in turn, underlies learning and memory formation [19], suggesting that spine turnover in the hippocampus directly reflects these processes [20,21]. Considering the low concentrations of anesthetics required to induce amnesia, these compounds are thought of being particularly effective on the hippocampus. One possible explanation of this sensitivity is the fact that a class of γ-aminobutyric acid receptors (GABARs), which is strongly modulated by some anesthetics, is predominantly expressed in the hippocampus [22,23]. Other anesthetics, such as ketamine, inhibit N-methyl-D-aspartate receptors (NMDARs) in a use-dependent manner and therefore may be particularly effective in inhibiting synaptic plasticity, required for the formation of episodic-like memories [24]. However, a systematic investigation of the effects of anesthetics on the hippocampus, bridging synaptic, network, and behavioral levels, is still lacking.

Here, using extracellular electrophysiological recordings and chronic 2-photon calcium and spine imaging in vivo in combination with behavioral analysis, we systematically assessed how CA1 network dynamics, synaptic structure, and memory performance are affected by 3 commonly used combinations of GAs: isoflurane (Iso), medetomidine/midazolam/fentanyl (MMF), and ketamine/xylazine (Keta/Xyl). We further measured CA1 network dynamics during wakefulness and natural sleep. Unlike sleep, all 3 GAs strongly reduced overall neuronal spiking compared to wakefulness. Moreover, opposite to what has been found in the neocortex [25–27], they decorrelated network activity, leading to a fragmented network state. However, the induced patterns of activity were highly distinct between the 3 different anesthetic

conditions and recovered to the preanesthetic status with disparate rates. Testing the effect of repeated anesthesia on spine dynamics revealed that Keta/Xyl, the condition which most strongly affected calcium activity, significantly reduced spine turnover, leading to an overall (over)stabilization of hippocampal synapses. In contrast, Iso and MMF mildly increased spine turnover. Finally, we show that the 2 anesthetic conditions which induce the strongest reduction and fragmentation of CA1 network activity, Keta/Xyl and MMF, negatively influenced hippocampus-dependent memory consolidation. On the other hand, Iso, which most closely resembled unperturbed sleep and wakefulness, did not impair memory consolidation, even when maintained over time periods matching the longer recovery phase of Keta/Xyl or MMF. Thus, different anesthetics, despite inducing a similar physiological state, strongly differ in their effects on synaptic stability, hippocampal network activity, and memory consolidation.

## Results

### Iso, Keta/Xyl, and MMF induce distinct patterns of network activity

Iso, Keta/Xyl, and MMF have distinct molecular targets and modes of action in the brain. We therefore hypothesized that electrical activity in the hippocampus might be uniquely altered by the 3 anesthesia strategies. To test this hypothesis, we investigated local field potentials (LFPs) and firing of individual neurons (single-unit activity, SUA) extracellularly recorded in the CA1 area of the dorsal hippocampus (dCA1) during wakefulness, followed by 45 minutes of anesthesia and 45 minutes of recovery (Fig 1A, S1A Fig). We found that the anesthetics differently affected population activity, inducing characteristic modulation of various frequency bands (Fig 1B). During wakefulness, LFP power in CA1 was highest in the theta (4 to 12 Hz) and low-gamma (40 to 60 Hz) frequency bands (S1B Fig). Exposure to 2% to 2.5% Iso led to a strong reduction of LFP power >4 Hz within the first 2 minutes, which was accompanied by complete loss of mobility of the animal (Fig 1C, S1B and S1C Fig). Similarly, MMF injection promptly decreased LFP power in the same frequency bands. In contrast, Keta/Xyl increased power across all frequencies during the first 10 minutes after injection, the most prominent effect being observed for activity at 5 to 30 Hz. This is consistent with previous reports, finding enhanced theta and low-gamma power in CA1 of rats under ketamine anesthesia [28]. The initial LFP power increase was followed by a gradual, significant decrease of 30 to 100 Hz activity (Fig 1C, S1B and S1C Fig).

It is widely accepted that, in the neocortex, GAs favor slow oscillations at the expense of faster ones [29]. To determine whether this is also the case in the hippocampus, we next asked how the investigated anesthetics affect slow network oscillations. Consistent with previous reports [30–32], Keta/Xyl strongly enhanced LFP power at 0.5 to 4 Hz throughout the entire recording period (Fig 1C and 1D, S1C Fig), but suppressed frequencies lower than 0.5 Hz. In contrast, Iso strongly augmented LFP power below 0.5 Hz, peaking at 0.1 to 0.2 Hz (Fig 1C and 1D, S1C Fig), whereas MMF induced no significant increase in the low-frequency regime. However, similar to Keta/Xyl, a significant reduction was present below 0.5 Hz, which persisted throughout the entire recording period (Fig 1C and 1D). Analysis of the power-law decay exponent (1/f slope) of the LFP power spectrum facilitates detection of noncanonical changes in LFP power, including aperiodic (non-oscillatory) components [33]. The 1/f slope has been hypothesized to track excitation/inhibition (E/I) balance [34,35] and is reduced in the cortex under anesthesia [36,37], indicating a shift toward inhibition. Considering the robust effects on LFP power that we reported, we reasoned that the 1/f slope might also be altered. Indeed, all anesthetics significantly decreased the 1/f slope, albeit with a different temporal profile. While the effect of Iso occurred within a few minutes, MMF and Keta/Xyl operated on a longer timescale (Fig 1E). Moreover, periods of activity were consistently and

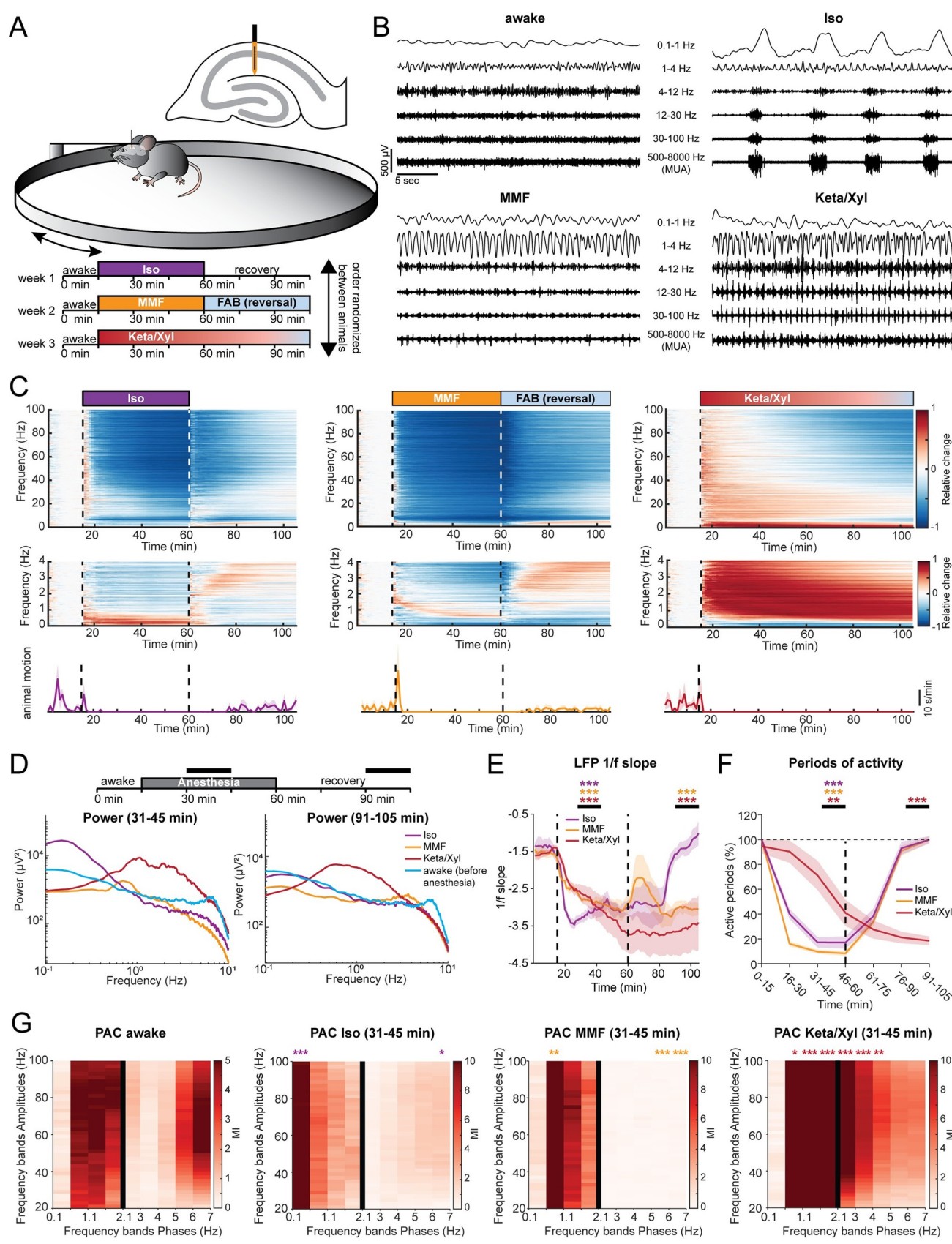

**Fig 1. LFP recordings in dorsal CA1 during wakefulness and anesthesia reveal distinct and complex alterations by Iso, Keta/Xyl, and MMF. (A)** Experimental setup. Extracellular electrical recordings in dorsal CA1 were performed in 4 head-fixed mice for 105 minutes, continuously. Each animal was recorded under all anesthetics as indicated in the scheme. Order of anesthetics was pseudo-randomized. **(B)** Characteristic LFP recordings during wakefulness and under 3 different anesthetics. **(C)** Color-coded heat maps depicting relative change (upper and middle panels) for LFP power and motion profiles (lower panels) for the 3 different anesthetic conditions. Upper panels display LFP power for 0–100 Hz frequency range, lower panels for 0–4 Hz. **(D)** Line plot displaying LFP power spectra for the 2 time periods indicated by horizontal black bars. For comparison, the 15-minute spectrum of the awake period before anesthesia induction is plotted in both graphs. Statistical differences are indicated in S1C Fig. **(E)** Line plot displaying the power-law decay exponent (1/f) of the LFP power spectrum for the 30–50 Hz range. Lines display mean ± SEM. **(F)** Line plot displaying the fraction of active periods compared to the preanesthetic wakeful state, in 15-minute bins throughout the entire recording duration. Lines display mean ± SEM. **(G)** Heat map displaying PAC for preanesthetic wakeful state (left) and for the indicated time periods during anesthesia. Different bin sizes (0.5 Hz and 1 Hz, separated by vertical black line) are used to resolve low- and high-frequency PAC. Vertical dashed lines in (C) and (E) indicate time points of anesthesia induction (Iso, MMF, and Keta/Xyl) and reversal (Iso and MMF only). Vertical dashed line in (F) indicates time point of anesthesia reversal (Iso and MMF only). Asterisks in (E) and (F) indicate significance of time periods indicated by black horizontal line compared to 15-minute period before anesthesia. Anesthetic conditions are color coded. Asterisks in (G) indicate significant differences compared to the corresponding frequency band during wakefulness. * $p < 0.05$, ** $p < 0.01$, *** $p < 0.001$, $n = 4$ mice. For full report of statistics, see S1 Table. All datasets of this figure can be found under https://github.com/mchini/Yang_Chini_et_al/tree/master/Stats_Dataset_(R)/datasets/Figure1_S1. FAB, flumazenil/atipamezole/buprenorphine; Iso, isoflurane; Keta/Xyl, ketamine/xylazine; LFP, local field potential; MMF, medetomidine/midazolam/fentanyl; PAC, phase-amplitude coupling.

strongly reduced immediately under Iso and MMF, but delayed by 30 minutes under Keta/Xyl (Fig 1F). These results indicate that all anesthetics shift the LFP to lower frequencies and tilt the E/I balance toward inhibition, albeit with different temporal profiles.

In contrast to Keta/Xyl anesthesia, Iso and MMF anesthesia can be efficiently antagonized. Removing the face mask is sufficient to antagonize Iso anesthesia, while antagonization of MMF anesthesia requires injection of a wake-up cocktail (flumazenil/atipamezole/buprenorphine, FAB) [38,39]. Moreover, 20 to 30 minutes after Iso withdrawal, animals regained motility and periods of silence in the LFP receded (Fig 1C and 1F). However, in contrast to post-Iso, LFP power did not fully recover after FAB, remaining significantly reduced at frequencies below 0.5 and above 30 Hz for the entire 45 minutes postanesthesia recording period (Fig 1C and 1D). In contrast, elevated LFP power in the 0.5 to 4 Hz band and reduction in active periods remained significant throughout the entire recording in the presence of Keta/Xyl. In line with these results, the 1/f slope promptly reverted to values similar to baseline after Iso discontinuation. In contrast, the recovery was only transitory and partial after MMF antagonization, and virtually absent for Keta/Xyl (Fig 1E), indicating that the E/I balance recovered only after Iso within 45 minutes.

Cross-frequency coupling between theta and gamma oscillations has been suggested to underlie information transfer in the hippocampus [40]. Given the strong decrease of theta power in the presence of Iso and MMF, we reasoned the phase modulation of the gamma rhythm could also be altered. To test this, we used phase-amplitude coupling (PAC) to measure whether the phase of slow LFP oscillations modulated the amplitude of the signal at a higher frequency. In line with previous results [41,42], a significant coupling between theta and gamma frequency bands, as well as between frequencies in the 1 to 2 Hz range and gamma, was present in the awake state (Fig 1G). Moreover, anesthesia strongly altered PAC. In accordance with the LFP power analysis, the coupling reached a maximum strength between the dominant slow-frequency oscillations induced by the various anesthetics (<0.5 Hz for Iso, approximately 1 Hz for MMF, and 0.5 to 4 Hz for Keta/Xyl) and gamma (Fig 1G). For all anesthetics, the range of phase-modulated amplitudes was wide, suggesting that the modulating phase corresponds to the identified slow-wave activity.

Taken together, these data show that all 3 GAs differently and persistently modulated the network oscillations in dCA1, a full recovery of activity being detected within 45 minutes only for Iso.

## Delayed recovery of neuronal spiking patterns after anesthesia

While the LFP provides information about general network states in the hippocampus, it is influenced by long-range activity and highly active regions in the vicinity of CA1 [43]. To assess the effects of GAs on CA1 neurons, we analyzed the spiking of individual units (56 to 72 units per animal, $n$ = 4 mice) before, during, and after each of the anesthetic conditions. All anesthetics significantly and rapidly (<1 minute) decreased spiking activity in CA1 neurons (Fig 2A and 2B, S2 Fig), with MMF leading to the most potent suppression, followed by Iso and Keta/Xyl. These alterations were generally present in all layers of CA1 (S2C Fig). Although the bulk spike rate was strongly reduced, the number of active neurons (see Methods) was only mildly affected (Fig 2C), reaching a significant reduction only with MMF. This observation suggests that anesthesia broadly reduces neuronal activity and does not modulate only a discrete subpopulation of neurons. Both firing rate and the number of active neurons recovered within 45 minutes after reversal for MMF and Iso (Fig 2A–2C, S2 Fig). As previously reported for non-rapid eye movement (NREM) sleep [44], we found a negative correlation between the anesthesia-induced reduction of firing rate and the firing rate in wakefulness (S2B Fig).

To investigate whether the rhythmicity of single neuron firing was affected similarly to the LFP, we analyzed the spectral properties of 1-ms binned SUA firing (i.e., power of SUA spike trains; for details, see Methods). In the presence of Iso, SUA power was consistently increased in the range between 0 and 0.5 Hz (Fig 2A and 2D, S2 Fig), in line with the strong modulation of LFP at 0.1 to 0.2 Hz. Of note, this effect did not vanish after Iso removal, suggesting that Iso has a long-lasting impact on firing rhythmicity. In contrast, and in line with its effects on the LFP, MMF generally reduced, albeit less strongly, SUA power, including the low frequencies. A significant reduction of SUA power was still present 45 minutes after antagonization in the 0 to 0.5 Hz band. Keta/Xyl, on the other hand, only showed a tendency toward reduced SUA power in the frequency band below 0.5 Hz, but increased SUA power significantly in the range between 0.5 and 4 Hz, consistent with its effect on the LFP (Fig 2D). This modulation was present throughout the entire recording. At higher frequencies, Iso led to a peak in the theta frequency range, similar to wakefulness (Fig 2E), yet it reduced the SUA power in the beta/gamma range. Keta/Xyl and MMF caused an overall reduction in SUA power at frequencies >5 Hz (Fig 2E). Thus, GAs differentially impair spiking rhythmicity. These changes appeared to follow similar dynamics than those in the LFP.

To confirm the synchrony between spikes and low-frequency oscillations, we calculated their pairwise phase consistency (PPC) [45]. When compared to pre-anesthesia, PPC values for the 0.1 to 0.5 Hz frequency band were augmented by Iso. Keta/Xyl increased coupling of spikes to the LFP between 0.5 and 1 Hz, whereas MMF showed a weak, but significant increase of coupling at frequencies below 1 Hz (Fig 2F).

Similar to the LFP, the SUA firing rate nearly fully recovered during the 45 minutes post-Iso (Fig 2A and 2B, S2 Fig), with even a slight, but significant increase at the end of the recording period. In contrast, after FAB-induced MMF reversal, CA1 spiking activity remained slightly reduced, reflecting the lack of LFP recovery. For Keta/Xyl, SUA remained suppressed during the entire recording period (Fig 2B). Strikingly, SUA power did not fully recover for any of the tested anesthetics (Fig 2E).

Taken together, we show that all investigated GAs caused a persistent and robust reduction of CA1 firing. Moreover, spiking during anesthesia was phase-locked to the GA-induced slow network oscillations.

## Iso, Keta/Xyl, and MMF reduce the number, amplitude, and duration of calcium transients

To monitor the population dynamics of CA1 neurons in the presence of different anesthetics, we imaged the same field of view (FOV) using the genetically encoded indicator GCaMP6f

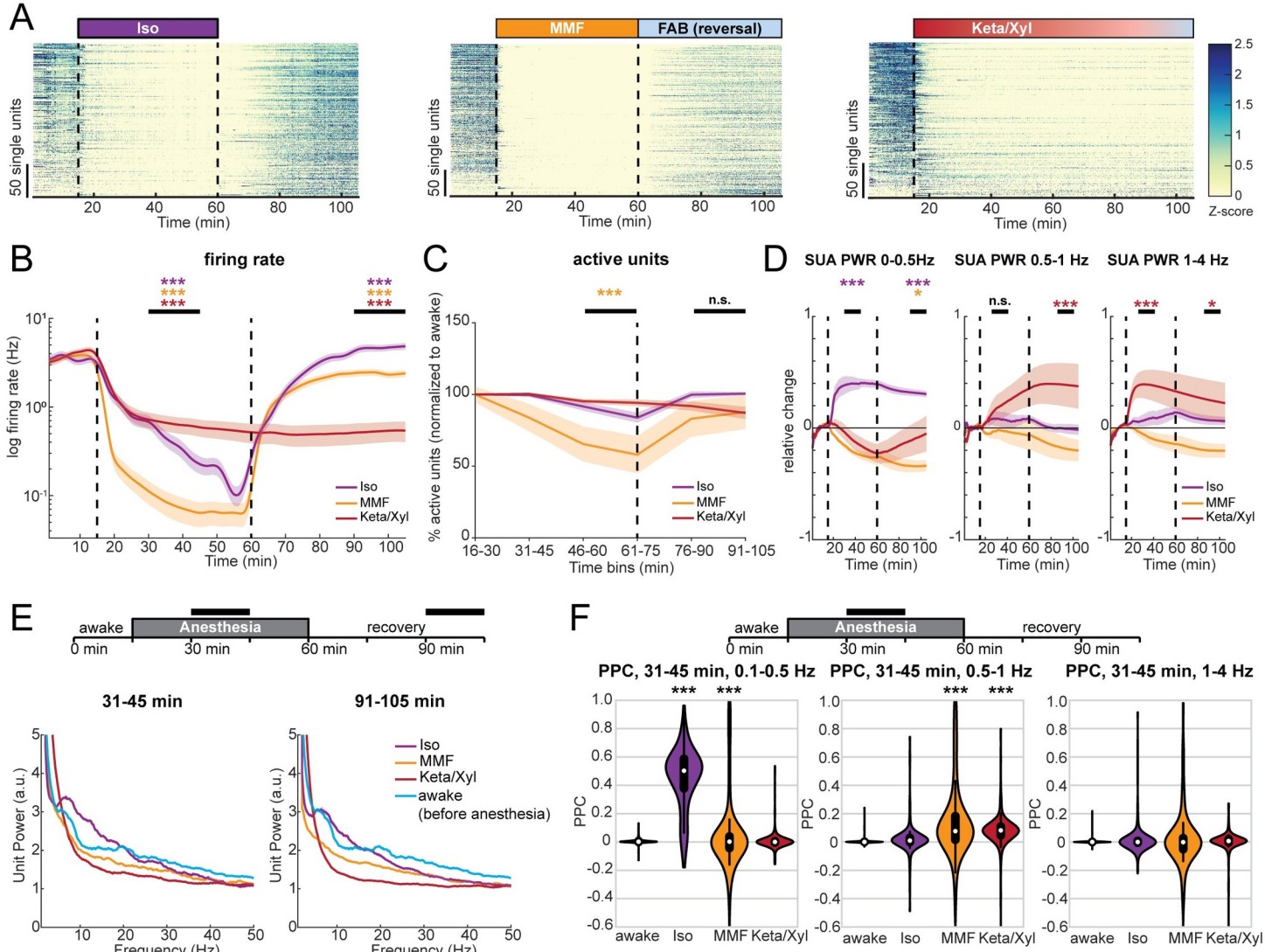

**Fig 2. SUA in dorsal CA1 is strongly reduced during anesthesia and remains significantly altered long after its termination. (A)** Raster plots of z-scored SUA for the 3 different anesthetic strategies in 4 mice. Units are sorted according to initial activity during wakefulness. **(B)** Line plot of SUA firing rate before, during, and after anesthesia induction. **(C)** Line plot displaying the fraction of active units compared to the preanesthetic wakeful state, for all 3 anesthetics in 15-minute bins throughout the entire recording duration. **(D)** Relative change of population firing rate power in the 0–0.5, 0.5–1, and 1–4 Hz frequency band. **(E)** Line plot displaying the normalized power spectra of population firing rate for the 2 time periods indicated by horizontal black bars. For comparison, the 15-minute spectrum for preanesthetic wakeful state is plotted in both graphs. **(F)** PPC at low frequencies in the same frequency bands as (D), for the indicated time points during anesthesia. White dots indicate median, and vertical thick and thin lines indicate first to third quartile and interquartile range, respectively. Colored lines in (B)–(D) display mean ± SEM. Vertical dashed lines in panels (A), (B), and (D) indicate time points of anesthesia induction (Iso, MMF, and Keta/Xyl) and reversal (Iso and MMF only). The vertical dashed line in (C) indicates the time point of anesthesia reversal (Iso and MMF only). Asterisks in (B)–(D) indicate significance of periods indicated by black horizontal line compared to period before anesthesia. Anesthetic conditions are color coded. Asterisks in (F) indicate significant differences to wakefulness. * $p < 0.05$, ** $p < 0.01$, *** $p < 0.001$, $n = 4$ mice. For full report of statistics, see S1 Table. All datasets of this figure can be found under https://github.com/mchini/Yang_Chini_et_al/tree/master/Stats_Dataset_(R)/datasets/Figure2_S2. FAB, flumazenil/atipamezole/buprenorphine; Iso, isoflurane; Keta/Xyl, ketamine/xylazine; MMF, medetomidine/midazolam/fentanyl; PPC, pairwise phase consistency; SUA, single-unit activity; SUA PWR, power of SUA spike trains.

[46] and systematically compared the activity of identified neurons during quiet wakefulness and in the presence of different anesthetics (Fig 3A).

First, we considered all active neurons in each condition and analyzed the average rate (i.e., the number of transients), amplitude, and duration (i.e., the decay constant) of calcium transients across all imaging sessions in 7 mice. In line with the results of SUA analysis (see

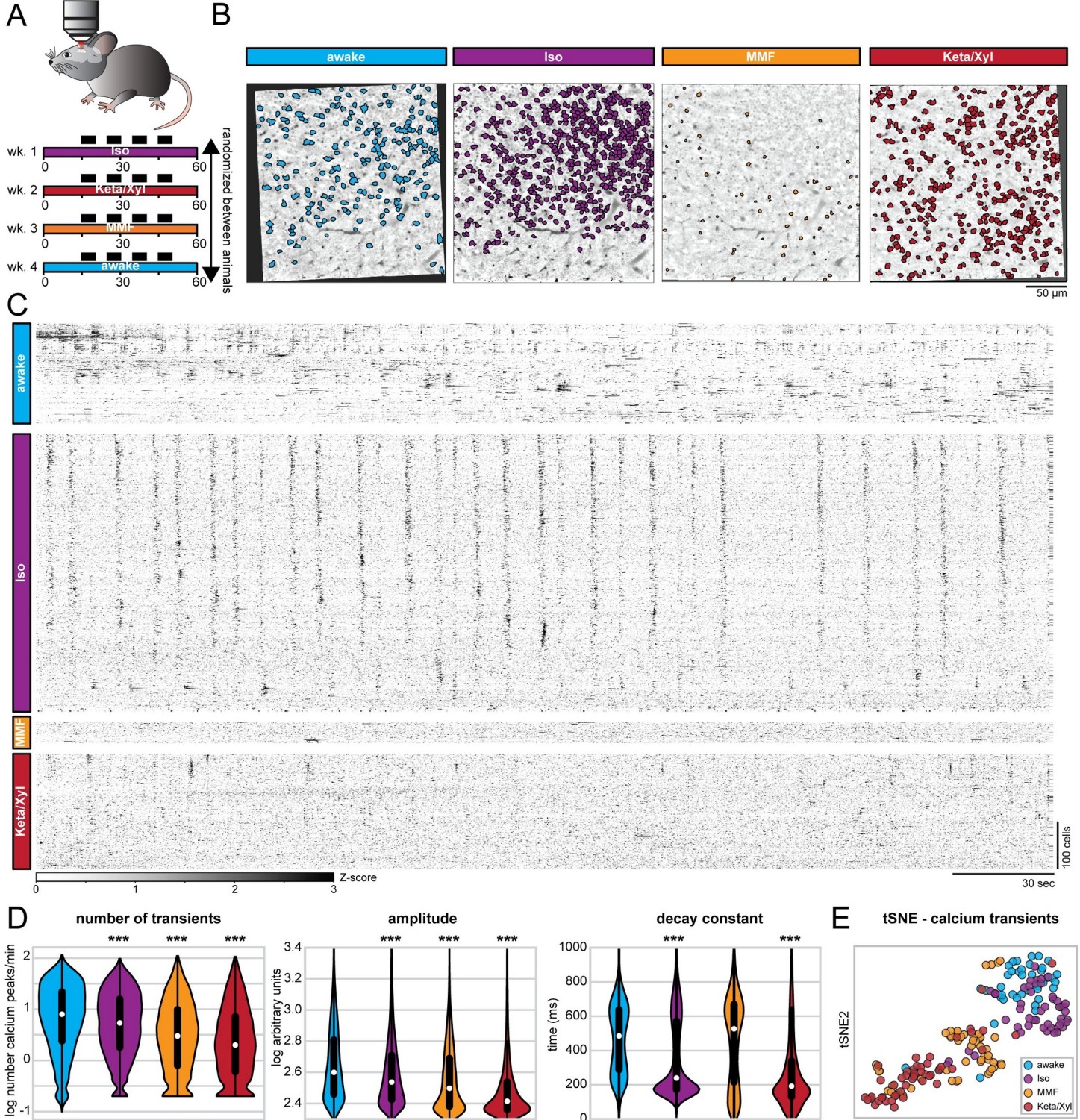

**Fig 3. Repeated calcium imaging in dorsal CA1 reveals distinct activity profiles for Iso, MMF, and Keta/Xyl. (A)** Experimental strategy for chronic calcium imaging of cellular activity in dorsal CA1. For each condition, 7 mice were imaged 4 times for 5 minutes, as indicated by black fields in the scheme. The order of imaging conditions was pseudo-randomized. **(B)** Time-averaged, 2-photon images of the same FOV in CA1 aligned to the Iso condition. ROIs of automatically extracted, active neurons are overlaid for each condition. **(C)** Raster plots of z-scored calcium transients in the same animal under different conditions. Traces are sorted by similarity. **(D)** Violin plots quantifying the number (left), amplitude (middle), and decay (right) of detected calcium transients. White dots indicate median, and vertical thick and thin lines indicate first to third quartile and interquartile range, respectively. **(E)** tSNE plot summarizing the average calcium transients properties. Each data point represents 1 recording

session. Asterisks in (D) indicate significant differences to wakefulness. *** $p < 0.001$. Note, to facilitate readability, only differences to wakefulness are indicated. For full report of statistics, see S1 Table. All datasets of this figure can be found under https://github.com/mchini/Yang_Chini_et_al/tree/master/Stats_Dataset_(R)/datasets/ Figure3_S3. FOV, field of view; Iso, isoflurane; Keta/Xyl, ketamine/xylazine; MMF, medetomidine/midazolam/fentanyl; ROI, region of interest; tSNE, t-distributed stochastic neighbor embedding.

Fig 2C), a large number of CA1 pyramidal neurons were active in the presence of all 3 GAs. Using extraction parameters that restricted the number of region of interests (ROIs) but maximized signal quality (see Methods), we obtained a median of 311 (min-max of 16 to 817) active neurons per FOV, for a total of one hundred and eighty-nine 5-minute recordings. All GAs significantly altered calcium dynamics in CA1 neurons, reducing the activity (Fig 3C and 3D), as previously shown for neuronal spiking (Fig 2B). Also in line with the effect on SUA (S2B Fig), the magnitude of the anesthesia-induced reduction of calcium transients was negatively correlated with the wakefulness calcium transients rate (S6D Fig). However, each condition could be characterized by a specific signature in their calcium dynamics. Iso yielded only a mild decrease of rate and amplitude, but a strong reduction of duration of calcium transients (Fig 3D). Consistent with effects on LFP and SUA, calcium transients showed a spectral peak between 0.1 and 0.2 Hz (S4 Fig). In contrast to Iso, MMF did not significantly affect the duration of transients but reduced their rate and amplitude when compared to wakefulness. Keta/Xyl anesthesia had the strongest effect on calcium transients, leading to a reduction of all 3 parameters compared to wakefulness (Fig 3D). Unlike for electrophysiological recordings, no spectral peak was present in calcium transients, most likely due to the strong suppression of calcium activity by Keta/Xyl. Considering all parameters, the 4 groups tended to segregate into clusters, one consisting mostly of recordings under Keta/Xyl and another one consisting of awake and Iso recordings. Most recordings under MMF clustered between these 2 groups (Fig 3E). Importantly, these findings were robust to changes in the signal extraction pipeline. Varying the threshold for calcium transient detection across a wide range of values did not affect the reported effects on rate and height of transients (S3B Fig). Further, conducting the same analysis on neuronal activity metrics that are independent of calcium transients detection (integral and standard deviation) or on dF/F calcium signals also yielded analogous results (S3C–S3E Fig).

## Iso, Keta/Xyl, and MMF distinctly modulate cellular calcium dynamics in individual neurons

One possible explanation for the distinct modes of calcium activity could be that each anesthetic condition recruits a unique set of neurons characterized by particular spiking properties. We tested this possibility by analyzing calcium transients in neurons that were active during all conditions (Fig 4A, S5 and S6 Figs). To obtain a sufficient number of active neurons, we extracted calcium transients using a lower quality threshold, accepting more neurons per recording (see Methods). In this manner, we obtained a median of 783 neurons per recording (min-max of 156 to 1,641). While this shifted the overall distribution of calcium parameters to lower values, the relative ratios between the 4 conditions remained the same, and the differences between anesthesia groups were preserved (S3F and S3G Fig). Also when considering only neurons that were active in all 4 conditions, rate as well as amplitude of calcium peaks were generally reduced under anesthesia, being lowest in the Keta/Xyl condition (Fig 4B and 4C). Compared to the whole dataset, differences in decay constant were less pronounced. The median decay constant strongly decreased for awake and MMF conditions, while it increased for Iso and Keta/Xyl. These results indicate that discrepancies between conditions generally decreased when considering only neurons active under all conditions.

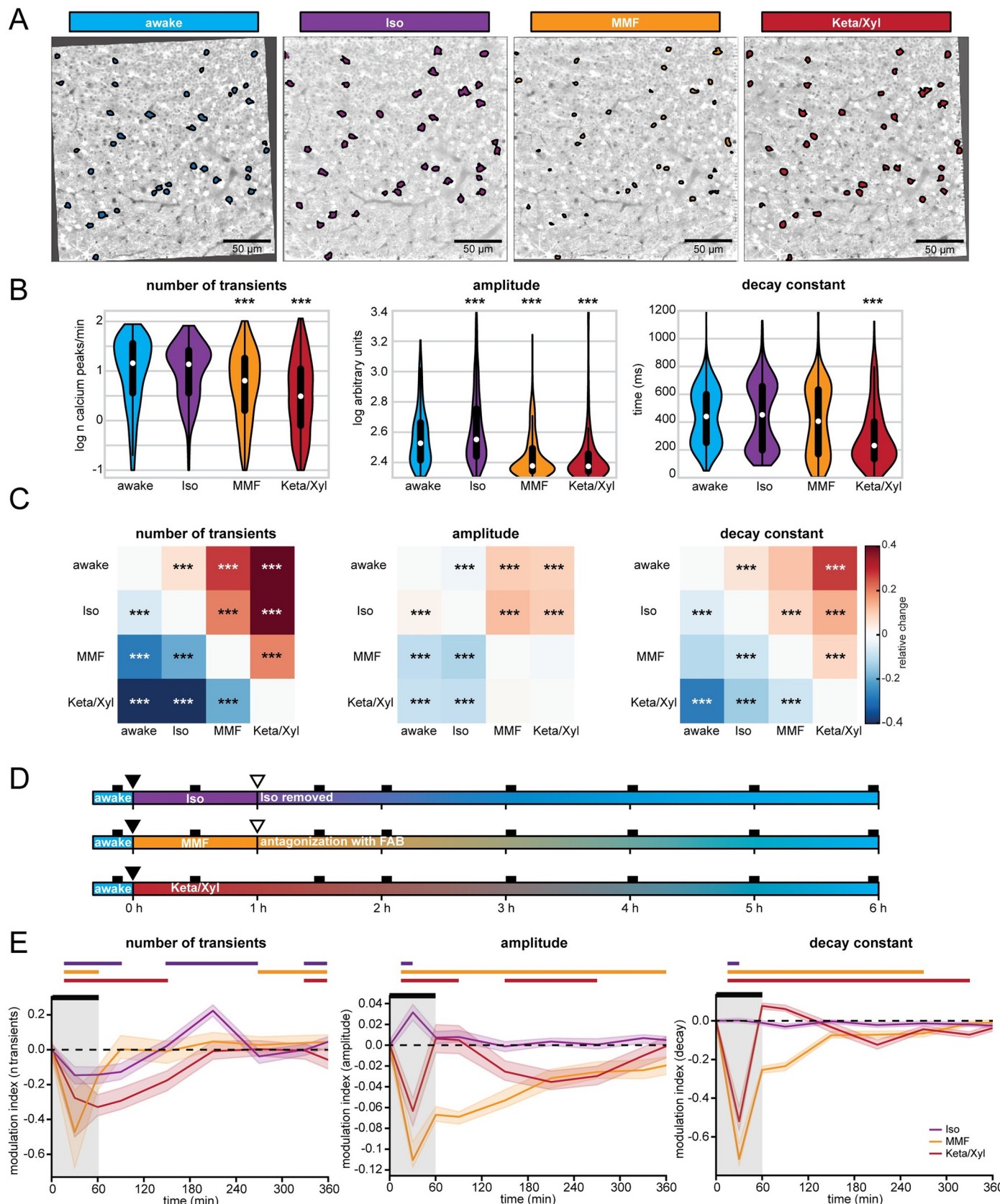

**Fig 4. Calcium activity profiles in neurons active during all conditions are similar between wakefulness and Iso. (A)** Two-photon time-averaged images of the same FOV in CA1, aligned to the Iso condition (same images as in Fig 3). ROIs show neurons active in each condition, allowing direct comparison of calcium transients in the same cells under different conditions. **(B)** Violin plots quantifying the number (left), amplitude (middle), and decay (right) of detected calcium transients. White dots indicate median, and vertical thick and thin lines indicate first to third quartile and interquartile range, respectively. **(C)** Heat maps displaying the relative change in the number (left), amplitude (middle), and decay (right) of calcium transients between neurons active in pairs of conditions (see also S6C Fig). **(D)** Schematic representation of long-term calcium imaging experiments to assess recovery from anesthesia. Black rectangles indicate imaging time points (up to 10 minutes duration each). Filled and open triangles indicate the start and end of the anesthesia period. **(E)** Line diagrams showing the relative change of the median number of calcium transients (left), their amplitude (middle), and decay constant (right) during anesthesia and recovery relative to the awake state before anesthesia induction. The black bar indicates the anesthesia period. Shaded, colored lines indicate 95% confidence interval. Note, Keta/Xyl anesthesia could not be terminated. The horizontal, colored lines indicate significant difference ($p < 0.05$) to awake time point ($t = 0$) for the respective condition. Asterisks in (B) and (C) indicate significant differences to wakefulness. *** $p < 0.001$. Note, to facilitate readability, only differences to wakefulness are indicated. For full report of statistics, see S1 Table. All datasets of this figure can be found under https://github.com/mchini/Yang_Chini_et_al/tree/master/Stats_Dataset_(R)/datasets/Figure4_S6. FOV, field of view; Iso, isoflurane; Keta/Xyl, ketamine/xylazine; MMF, medetomidine/midazolam/fentanyl; ROI, region of interest.

The relatively low number of neurons active in all 4 conditions (335 neurons) limited the statistical analysis. Therefore, we compared neurons that were active in any 2 combinations of conditions (S6C Fig). This analysis further corroborated the similarity of neurons active during wakefulness and Iso anesthesia (Fig 4C, S6C Fig). Rate, amplitude, and duration of calcium transients were most similar between wakefulness and Iso compared to the other GAs. In contrast, neurons active during wakefulness and either Keta/Xyl or MMF showed decreased rate, amplitude, and duration under anesthesia, with Keta/Xyl causing the strongest phenotype (S6C Fig). Overall, this indicates that anesthetics influence the firing properties of hippocampal neurons. However, the magnitude and direction of these effects vary considerably. On the one hand, Iso anesthesia has the mildest effect, and it most likely arises from distinct neuronal populations being active in the 2 conditions (wakefulness versus Iso anesthesia), as the firing properties of cells that are active in both are barely affected (Fig 4B and 4C). On the other hand, the strong effects of MMF and Keta/Xyl on all calcium parameters in the same cells indicate that different anesthetics directly alter the firing properties of individual neurons. Thus, alterations in firing properties of neuronal populations (e.g., SUA, Fig 2B and 2D) are not solely explainable by different subpopulations of neurons being active between awake and anesthesia.

## Population activity recovers with different temporal dynamics after Iso, Keta/Xyl, and MMF

The LFP recordings showed that network activity remained altered for 1.5 hours after Keta/Xyl injection, but also after antagonization of MMF, while most aspects returned to preanesthetic conditions during 45 minutes after Iso removal. To assess network effects of the different anesthetics on a longer timescale, we used repeated calcium imaging during 6 hours after anesthesia onset and 5 hours after Iso termination and MMF antagonization (Fig 4D). In line with our previous results, the number of calcium transients was strongly reduced 30 minutes after MMF or Keta/Xyl injection, while the reduction had a lower magnitude for Iso. Similarly, MMF and Keta/Xyl most strongly reduced the amplitude and duration of calcium transients, while Iso mildly increased amplitude without affecting the decay constant (Fig 4E).

Confirming the dynamics monitored by LFP recordings in vivo, recovery from Iso anesthesia was fast, and only the rate of transients mildly changed during the hours after removing the mask. In contrast, after Keta/Xyl injection, amplitude and duration of transients were altered throughout the following 6 hours, while the reduction of the calcium transients rate was not reverted until up to 4 hours later. Recovery to the preanesthetic state was even slower after MMF/FAB. Despite antagonization of MMF anesthesia with FAB, calcium transients remained disturbed for up to 6 hours. Thus, the different anesthetics not only induce unique alterations of CA1 network dynamics, but also show different recovery profiles (S6E Fig).

## Anesthesia decorrelates hippocampal activity

Calcium imaging studies in the visual cortex of ketamine-anesthetized rats [27] and Iso-anesthetized mice [25] showed that anesthesia increases the overall pairwise correlations between firing neurons and, consequently, induces more structured patterns of activity. While neocortical L2/3 cells typically show a high degree of local interconnectivity [47], this is not the case for CA1, where pyramidal cells receive their main excitatory input from CA3 and entorhinal cortex and send their efferents to subiculum and extrahippocampal areas [9]. Another difference between neocortex and hippocampal CA1 area is that the neocortex receives strong direct input from primary thalamus, which is a major source for slow oscillations during anesthesia-induced unconsciousness and sleep [1,48,49]. In comparison to neocortex, hippocampus shows different patterns of activity, including sharp waves, which are generated intrinsically in the hippocampus, likely originating in CA3 [50]. To investigate whether these differences cause a different impact of anesthesia on the population activity in CA1 when compared to the neocortex, we analyzed the dynamical structure of population activity using both calcium imaging and SUA of extracellular recordings in vivo. First, we analyzed Fisher-corrected Pearson pairwise correlation between neuropil-corrected raw fluorescence traces. We found that both correlation and anticorrelation were highest in animals during quiet wakefulness (Fig 5A and 5B). In particular, the awake condition had a higher proportion of correlation coefficients both in the first as well as in the fourth quartile of the entire distribution and, accordingly, higher absolute correlation values (Fig 5B, S7A Fig). Similar to the firing properties (SUA, Fig 2), Iso induced the milder changes, whereas Keta/Xyl caused the strongest phenotype. This relationship was preserved in neurons active during all conditions (S7B Fig), indicating that anesthesia generally reduces correlated activity between neurons and that this effect is not attributable to the activity of particular neuronal subpopulations. Moreover, these effects were not influenced by the distance between the pair of neurons whose correlation was quantified (Fig 5C). These findings highlight the major differences between the anesthesia-induced effects on neuronal coupling in hippocampal CA1 and neocortex. In accordance with the anatomy of CA1, the correlation between pairs of neurons was only mildly affected by the distance between them, with or without anesthesia. Not only were neurons less highly correlated to each other under anesthesia, but also their coupling to the whole population activity [51] was reduced as well. The proportion of neurons with population coupling in the fourth quartile of the entire distribution was highest for awake and most strongly reduced under Keta/Xyl and MMF, while Iso showed only mild effects (Fig 5D).

To further relate the calcium imaging data to extracellular recordings of neuronal firing, we carried out an analogous analysis on SUA. To avoid the confounding effect of firing rate, we quantified the correlation between pairs of neurons using the spike time tiling coefficient (STTC) [52], a measure that is largely insensitive to variations of the firing rate (see Methods). To be consistent with the calcium data, we quantified correlations within 1 second, a timescale of the same magnitude as the decay constant used to extract calcium signals (700 ms). This analysis confirmed that all anesthetics decorrelated neuronal activity (Fig 5E). This effect was still present, albeit less pronounced, using an integration window of 10 ms, which is closer to the duration of action potentials (S7C Fig). Overall, the decorrelation was milder under Iso anesthesia and stronger under Keta/Xyl and MMF. Thus, all 3 GAs decorrelated calcium transients and spiking activity in the CA1 area, with MMF and Keta/Xyl inducing the most prominent effects.

## Anesthesia fragments temporal and spatial structure of hippocampal activity

The decorrelation of neuronal activity during anesthesia suggests that GAs might impact the spatial and temporal organization of CA1 neuronal ensembles (see Fig 5A). To test this

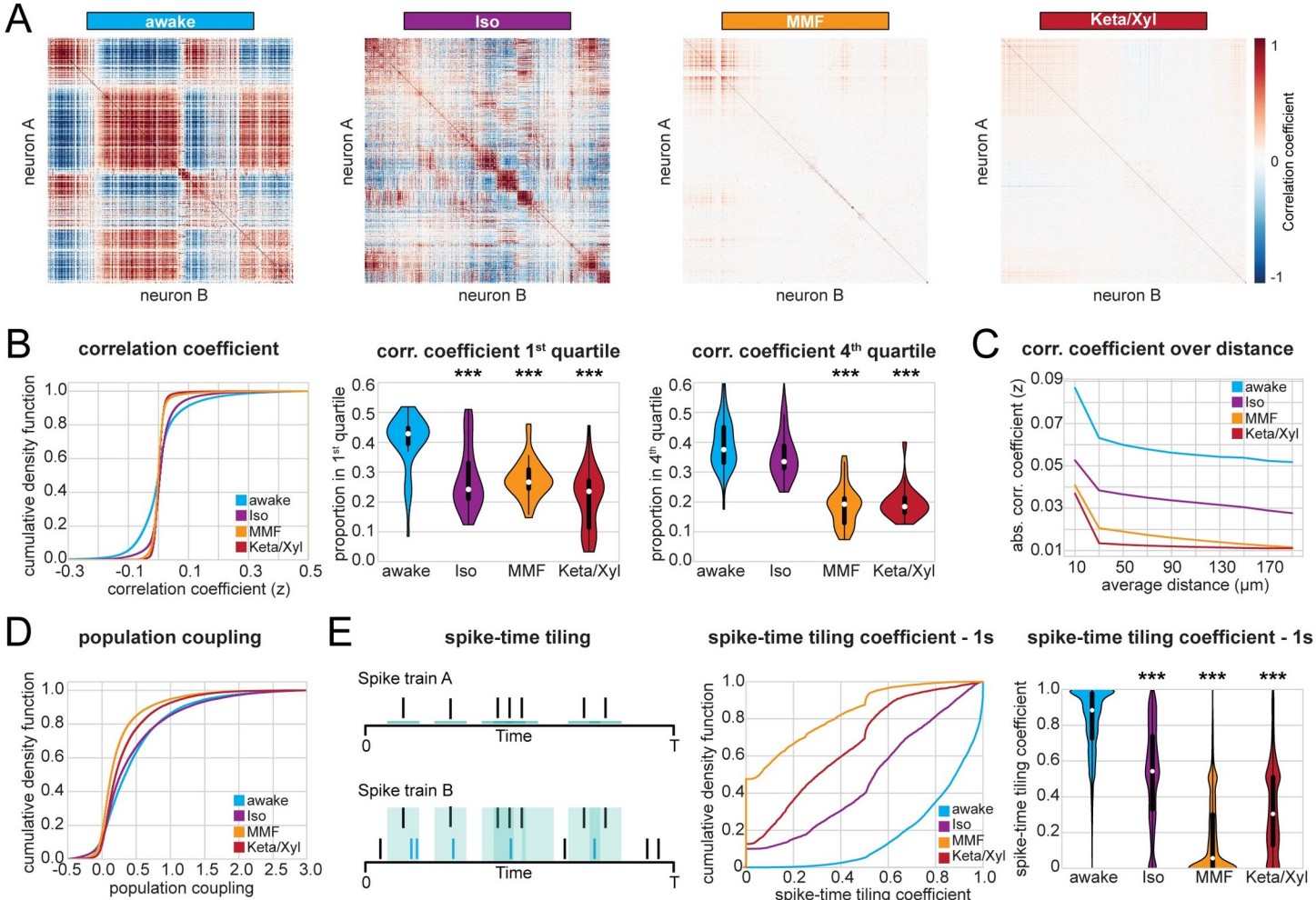

**Fig 5. Correlation analysis of CA1 calcium activity and SUA shows decorrelation under anesthesia.** (A) Heat maps displaying representative correlation matrices of calcium activity between pairs of neurons during wakefulness and the 3 different anesthetic conditions in the same animal. Matrices are sorted by similarity. (B) Left: Line plot displaying cumulative distribution of Fisher-corrected Pearson correlation coefficients between pairs of neurons (calcium imaging). Center: violin plot displaying the proportion of pairs found in the first (most negative) and fourth (most positive) quartile of the distribution. (C) Line plot displaying the absolute pairwise correlation coefficients over distance (calcium imaging, 25-micrometer bins). (D) Line plot displaying the cumulative distribution of population coupling (calcium imaging). (E) Quantification of correlation between pairs of extracellularly recorded single units using the STTC. Left: Schematic illustration of the STTC quantification. Center: cumulative distribution of the STTC with a 1,000-ms integration window. Right: violin plot quantifying the STTC. In violin plots, white dots indicate median, and vertical thick and thin lines indicate first to third quartile and interquartile range, respectively. Asterisks in (B) and (E) indicate significant differences to wakefulness. *** $p < 0.001$. Note, only differences to wakefulness are indicated. For comparison between conditions, see S1 Table. All datasets of this figure can be found under https://github.com/mchini/Yang_Chini_et_al/tree/master/Stats_Dataset_(R)/datasets/Figure5_S7. Iso, isoflurane; Keta/Xyl, ketamine/xylazine; MMF, medetomidine/midazolam/fentanyl; STTC, spike time tiling coefficient; SUA, single-unit activity.

hypothesis, we analyzed the same number of active neurons for each condition, since a different number of neurons in each condition potentially influences the number and size of detected clusters. First, we monitored the impact of GAs on the temporal structure of CA1 activity. We defined the number of clusters identified by principal component analysis (PCA) as the number of components that were needed to explain 90% of the variance. Moreover, we assessed the power-law slope of variance explained over the first 100 components (Fig 6A). Both methods led to a larger number of clusters and a flatter power-law slope for anesthesia when compared to wakefulness (Fig 6A). Further corroborating these findings, both t-distributed stochastic neighbor embedding (tSNE) dimensionality reduction and affinity propagation

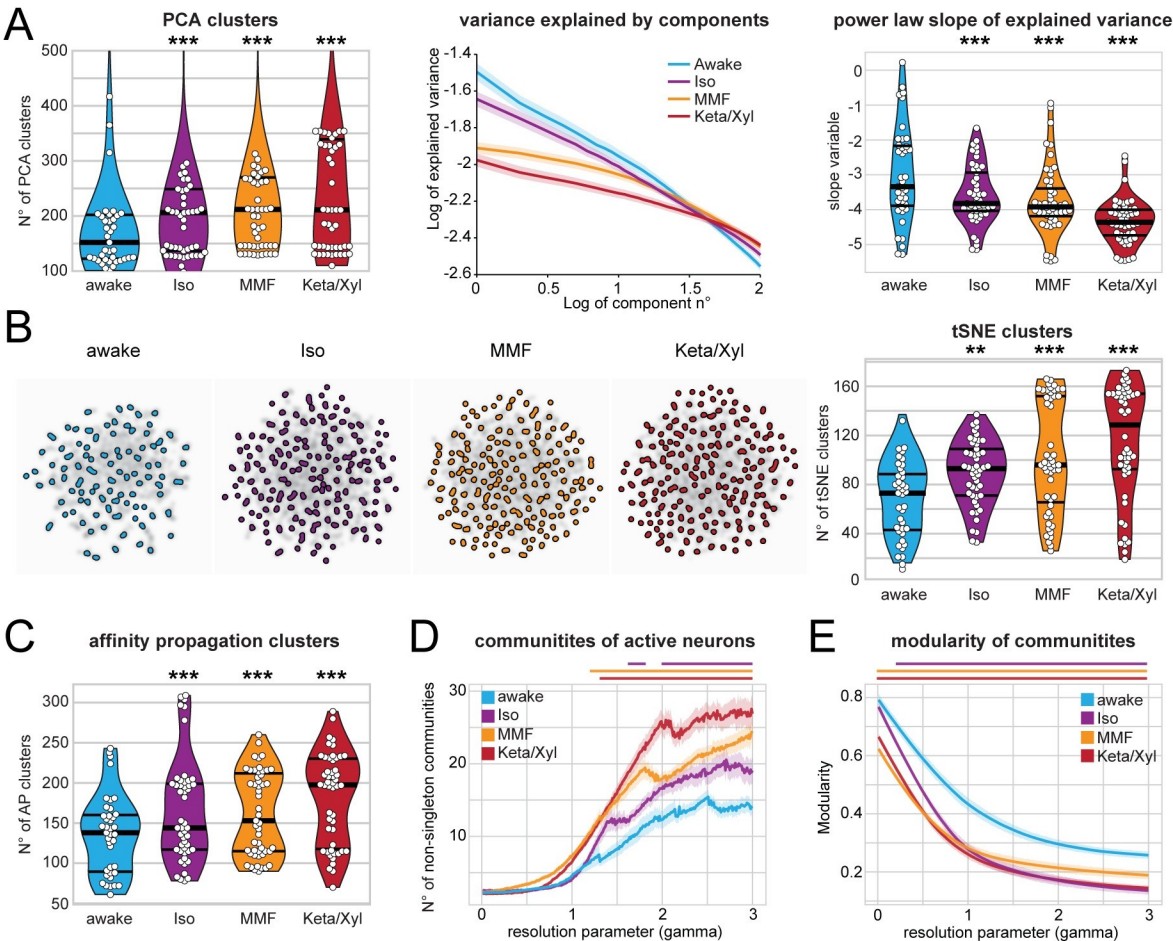

**Fig 6. Calcium activity in CA1 is temporally and spatially fragmented during anesthesia. (A)** Left: violin plot quantifying the number of PCA clusters during wakefulness or anesthesia, as indicated. Middle: log-log line plot displaying the variance explained by the first 100 components for each condition. Right: violin plot quantifying the power-law slope of the variance explained by the first 100 components for each condition. **(B)** Left: tSNE plots of network events recorded in the same animal under the 4 indicated conditions. Right: Violin plot quantifying the number of tSNE clusters obtained from calcium recordings during the 4 different treatments. **(C)** Violin plot quantifying the number of clusters obtained by AP from calcium recordings during the 4 different treatments. **(D)** and **(E)** Line plots quantifying the number of detected communities and the modularity of the detected communities with the resolution parameter gamma ranging from 0 to 3. Horizontal lines in violin plots indicate median and first to third quartile. Asterisks in (A)–(C) indicate significant differences to wakefulness. $^{**}$ $p < 0.01$, $^{***}$ $p < 0.001$. Horizontal lines above plots in (D) and (E) indicate significant difference to wakefulness. Anesthetic conditions are color coded. Note, only differences to wakefulness are indicated. For comparison between conditions, see S1 Table. All datasets of this figure can be found under https://github.com/mchini/Yang_Chini_et_al/tree/master/Stats_Dataset_(R)/datasets/Figure6_S8. AP, affinity propagation; Iso, isoflurane; Keta/Xyl, ketamine/xylazine; MMF, medetomidine/midazolam/fentanyl; PCA, principal component analysis; tSNE, t-distributed stochastic neighbor embedding.

(AP) clustering (see Methods) also revealed a larger number of clusters for anesthesia compared to wakefulness (Fig 6B and 6C). These observations indicate that activity is less structured under anesthesia. In line with previous results, Iso had the weakest effect, whereas Keta/Xyl consistently induced the most pronounced phenotype. Analysis of the deconvolved calcium traces led to comparable results (S8A and S8B Fig). These findings support the idea that GAs fragment the hippocampal network into a more diverse repertoire of microstates.

Second, we tested whether anesthesia disrupted the spatial structure of hippocampal activity, employing a modularity maximization approach [53,54] designed to detect internally densely connected communities (modules). To allow detection of modules at varying sizes, we carried out our analysis while varying a resolution parameter (gamma) and thus focusing on

different spatial scales. Using this approach, we showed that GAs increase the number of detected communities over a wide range of resolution parameter values (Fig 6D). Moreover, the modularity of these communities was lower than in wakefulness (Fig 6E). These results indicate that anesthesia results in a more fractured network with, on average, smaller and less coherent communities. A multiresolution approach [55] followed by the selection of partitions based on hierarchical consensus clustering yielded similar results (S8C Fig). Among GAs, Iso induced the mildest phenotype, whereas Keta/Xyl had the most prominent effects. Thus, GAs not only decorrelate hippocampal activity, but also consistently fragment both its temporal and spatial structure.

## Network alterations during sleep are less pronounced compared to anesthesia

Altered CA1 activity under anesthesia may affect synaptic function and memory processing. A naturally occurring form of unconsciousness is sleep, which is required for network processes involved in memory consolidation [49,56]. To decide whether the network perturbations described above resemble those naturally occurring during sleep, we first monitored CA1 activity by recording the LFP and spiking together with animal motion and the neck-muscle electromyography (EMG) in head-fixed mice (S9A Fig). We classified the signal into 30-second long epochs of wake, rapid eye movement (REM) and NREM sleep. Further, a certain fraction of epochs, which we labeled as "uncertain," could not be reliably classified into any of the previous 3 categories (see Methods for details). Given that the behavioral attribution of these epochs is uncertain and difficult to interpret, we excluded them from further analysis. The animals spent most of their sleeping time in the NREM phase, with only short periods of intermittent REM sleep (Fig 7A and 7I). The LFP showed enhanced theta power during REM phases, while the power at low frequencies was broadly increased during NREM sleep (Fig 7B and 7C). Compared to anesthesia (Fig 1), these changes in the LFP both during REM and NREM phases were modest. Along the same line, the 1/f slope during NREM and REM sleep slightly decreased, indicating a small reduction of the E/I balance that had a significantly lower magnitude than the perturbation induced by GAs (Fig 7D). Furthermore, the SUA rate was slightly reduced only in REM sleep (Fig 7E and 7F), in contrast to all anesthetics, that strongly suppressed firing (Fig 2). As previously reported [44], and similarly to the effect of GAs, we detected a small but significant negative correlation between the NREM-induced reduction of firing rate and the wakefulness firing rate, whereas the effect failed to reach statistical significance for REM sleep alone (Fig 7G). Moreover, NREM sleep induced a small reduction of pairwise correlation between pairs of neurons, as measured by the STTC with an integration window of 1 second.

To additionally investigate the effect of sleep on hippocampal activity, we used the abovementioned recordings to train a machine learning algorithm to classify wakefulness, NREM, and REM sleep from eye videography images alone (S9 Fig) [57] (see Methods for details). In line with previous results, we were able to reliably distinguish wakefulness and NREM sleep (4-fold cross-validation accuracy >85%), whereas REM classification was less precise (4-fold cross-validation accuracy approximately 30%). This classifier was then used to predict the physiological state of mice from which we recorded calcium transients in CA1 neurons. In the calcium imaging dataset, sleep was dominated by the NREM phase, and only 17 minutes of REM sleep could be detected in a total of 864 minutes (Fig 7I and 7J). Given the limited amount of detected REM sleep, its effects on hippocampal calcium activity should be interpreted with caution. As reported for LFP data, NREM only mildly reduced the rate of calcium transients, whereas REM sleep induced a small increase. In contrast, both NREM and REM

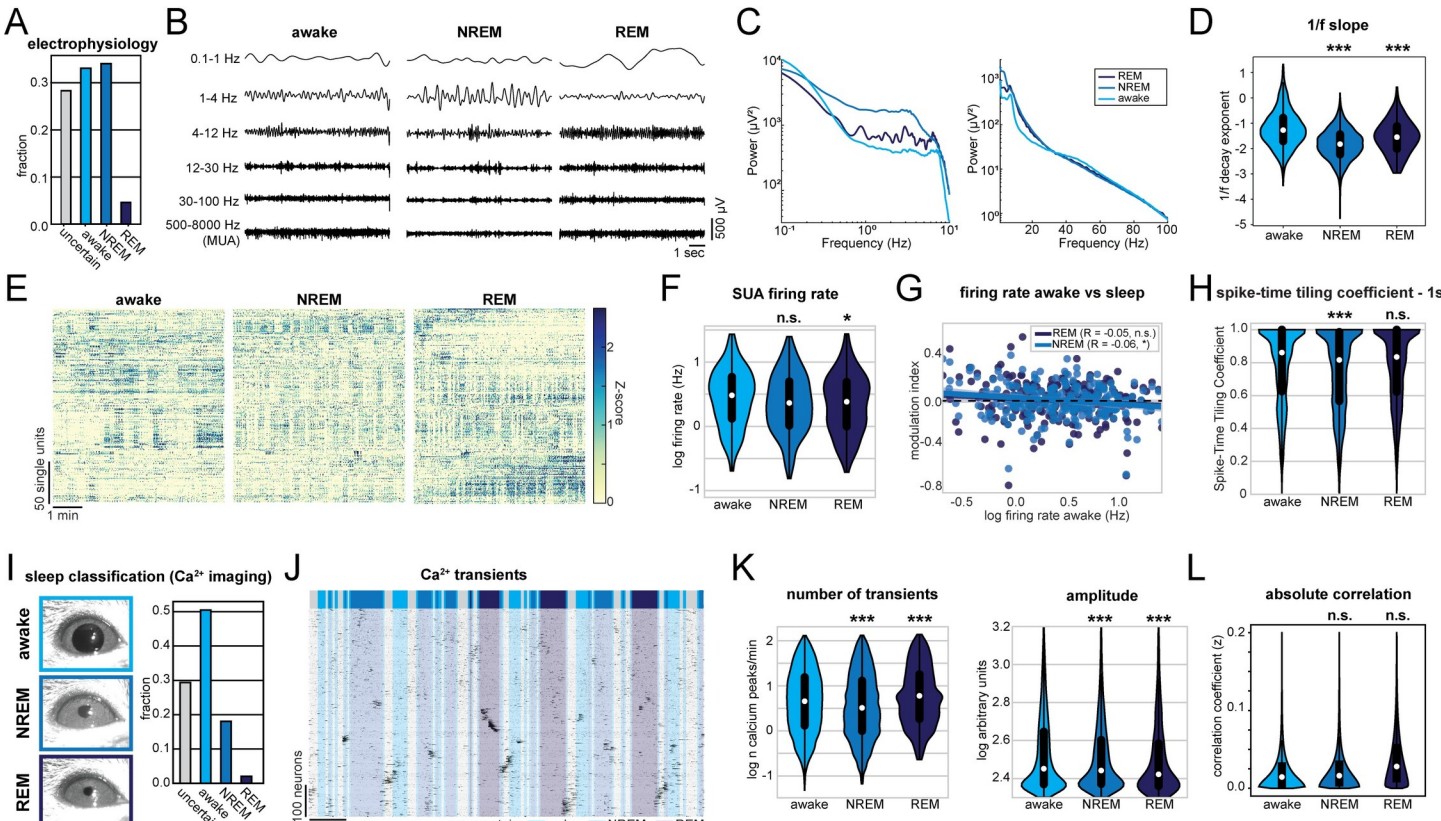

**Fig 7. Sleep alters CA1 activity in a similar way to anesthesia but with a lower magnitude.** (A) Classification of activity states during electrical recordings. (B) Characteristic LFP recordings during wakefulness, NREM, and REM sleep. (C) Line plot displaying LFP power spectra for the indicated activity states. (D) Violin plot displaying the power-law decay exponent (1/f) of the LFP power spectrum. (E) Raster plots of z-scored SUA for the 3 different activity states in 4 mice. Units are sorted according to initial activity during wakefulness. (F) Violin plot showing SUA firing rate. (G) Scatter plot showing modulation of SUA firing rate during NREM (light blue) and REM sleep (dark blue) with respect to activity during wakefulness. (H) Violin plot quantifying the STTC. (I) Classification of activity states during CA1 calcium imaging based on eye videography. (J) Raster plots of z-scored calcium transients in an example recording of 1 animal transiting between wakefulness and sleep. Traces are sorted by similarity. (K) Violin plots quantifying the number (left) and amplitude (right) of detected calcium transients. (L) Violin plots quantifying absolute pairwise correlation of all recorded neurons. White dots indicate median, and vertical thick and thin lines indicate first to third quartile and interquartile range, respectively. * $p < 0.05$, ** $p < 0.01$, *** $p < 0.001$ w.r.t. to wake state, $n = 3–7$ mice. For full report of statistics, see S1 Table. All datasets of this figure can be found under https://github.com/mchini/Yang_Chini_et_al/tree/master/Stats_Dataset_(R)/datasets/Figure7_S9. LFP, local field potential; NREM, non-rapid eye movement; REM, rapid eye movement; STTC, spike time tiling coefficient; SUA, single-unit activity.

sleep caused a small reduction in transient amplitude (Fig 7K). Further, we did not detect an effect of the sleep state on absolute pairwise correlations (Fig 7L).

In conclusion, sleep and GAs similarly affect the CA1 activity. However, the magnitude of effects was much smaller for sleep than for GAs. Both NREM and REM states were more similar to wakefulness than to the anesthetic state. Compared to the 3 different anesthetics, sleep had the closest resemblance to Iso. Thus, among the 3 different anesthetics, network alterations under Iso deviate the least from natural states such as wakefulness and sleep.

## Repeated anesthesia alters spine dynamics in CA1

The impact of Iso, MMF, and Keta/Xyl on CA1 activity might alter spine dynamics at CA1 pyramidal neurons. This issue is of critical relevance, since GAs disrupt activity patterns during development [58] also involving alteration of synaptic connectivity [59–61], but less is known about the impact of GAs on hippocampal synaptic structure during adulthood. So far, spine dynamics in hippocampus were only investigated under anesthesia, lacking comparison

to the wake state. Moreover, the reported turnover rates varied strongly between studies [21,62,63]. Thus, it is unknown how repeated anesthesia in itself affects spine stability.

We repeatedly imaged the same basal, oblique, and tuft dendritic segments of CA1 pyramidal neurons under all 4 conditions (5 times per condition, every 4 days), interrupted by a

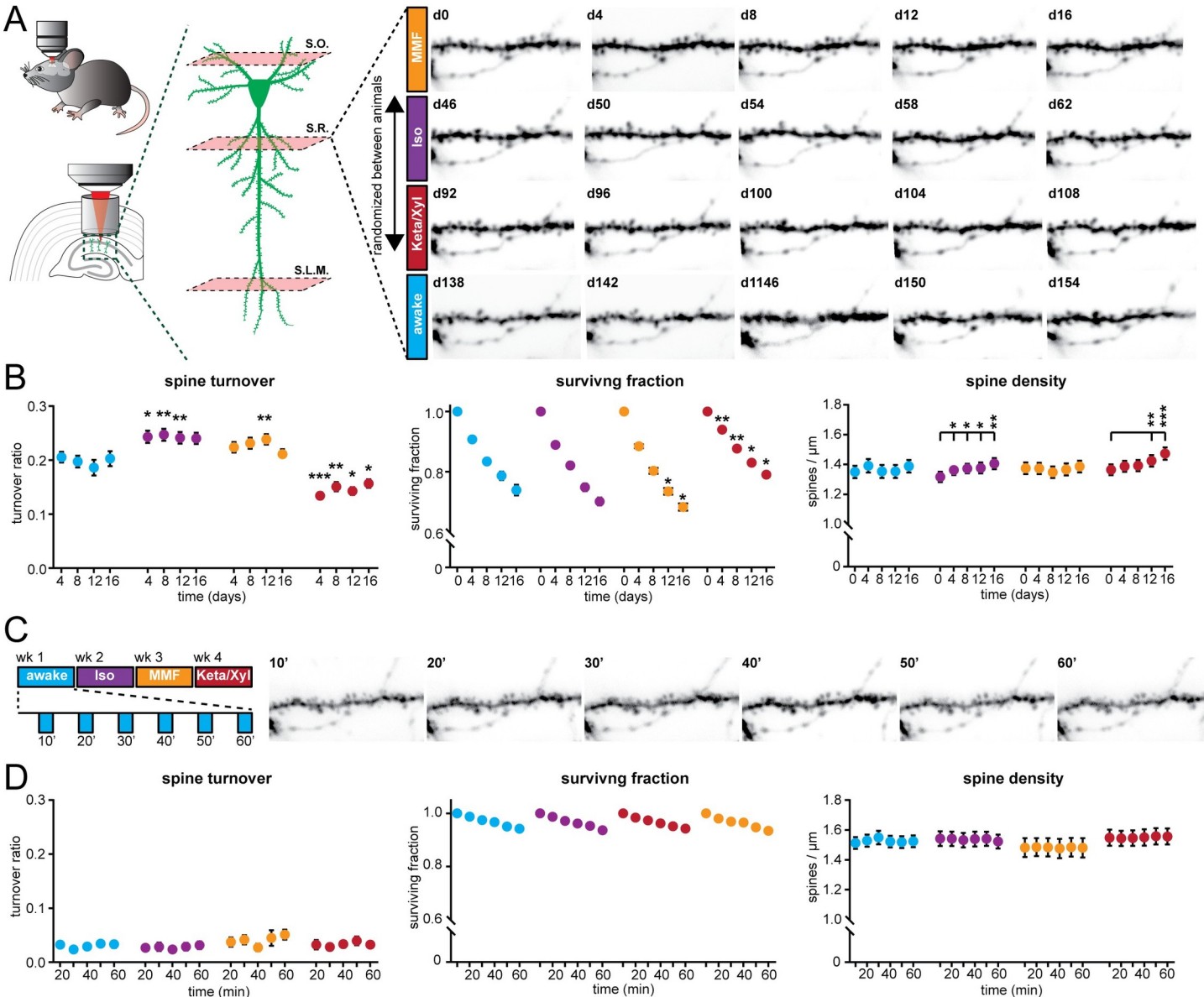

**Fig 8. Spine turnover at CA1 pyramidal neurons is distinctly altered by repeated application of Iso, MMF, and Keta/Xyl. (A)** Left: Schematic illustration of in vivo spine imaging strategy. In each animal, spines were imaged on basal dendrites located in S.O., oblique dendrites in S.R., and tuft dendrites in S.L.M. Right: Example showing an oblique dendrite in S.R. imaged chronically during all conditions. The order of anesthetic treatments was pseudo-randomized between mice (see S10A Fig). **(B)** Dot plots showing quantification of spine turnover (left), spine survival (middle), and spine density (right) under the 4 indicated treatments. Note that spines were imaged on the same dendrites across all conditions. Dots indicate mean ± SEM. Asterisks indicate significant differences to wakefulness in the left and middle panel. In the right panel, asterisks denote significant changes within each treatment compared to day 0. * $p < 0.05$, ** $p < 0.01$, *** $p < 0.001$. **(C)** Imaging of acute spine dynamics during 4 different conditions. Left: schematic of the experimental timeline. Right: example of dendrite imaged during wakefulness in 10-minute intervals (same dendrite as in A). **(D)** Dot plots showing quantification of acute spine turnover (left), spine survival (center), and spine density (right) under the 4 indicated treatments. Dots indicate mean ± SEM. For full report of statistics, see S1 Table. All datasets of this figure can be found under https://github.com/mchini/Yang_Chini_et_al/tree/master/Stats_Dataset_(R)/datasets/Figure8_S10. Iso, isoflurane; Keta/Xyl, ketamine/xylazine; MMF, medetomidine/midazolam/fentanyl; S.L.M., stratum lacunosum moleculare; S.O., stratum oriens; S.R., stratum radiatum.

30-day recovery period between conditions (Fig 8A, S10A Fig). To rule out time effects, we pseudo-randomized the order of anesthetics (S10A Fig). During wakefulness, without any anesthesia in between, the turnover ratio of spines on all dendrites was on average 18.6% to 20.5% per 4 days. This turnover ratio was stable and did not change systematically over successive imaging sessions (Fig 8B). Notably, all anesthetics affected spine turnover. Both MMF and Iso anesthesia mildly increased the turnover ratio compared to wakefulness (21.1% to 23.8% for MMF and 24.0% to 24.7% for Iso). Iso did not alter the surviving fraction of spines. Together with the significant increase in spine density over time (Fig 8B), these results indicate that the elevated turnover ratio was due to a rise in the gained fraction of spines (S10B Fig). In contrast, MMF led to a slight increase in the fraction of lost spines (S10B Fig) and correspondingly, slightly decreased the surviving fraction compared to wakefulness. Spine density did not change over time. Keta/Xyl anesthesia showed the strongest effect on spine turnover (13.4% to 15.7%), which was opposite to MMF and Iso, and therefore significantly lower, rather than higher, compared to the awake condition (Fig 8B). This lower turnover ratio was accompanied by a higher surviving fraction and an increase in density with time (Fig 8B). Consistently, the fraction of lost spines was most strongly reduced (S10B Fig). Thus, Keta/Xyl anesthesia resulted in marked stabilization of existing spines and a reduction in the formation of new spines, indicative of a significant effect on structural plasticity. These effects were present on basal dendrites of S.O., oblique dendrites in S.R., and tuft dendrites in S.L.M., albeit with different magnitude. Under Keta/Xyl, the strongest impact on spine density was present in S.L.M., while turnover was most strongly reduced in S.O. Also, the increased spine turnover seen under Iso and MMF was most pronounced in S.O. (S10D Fig).

To rule out that the age of the animal-influenced spine dynamics in the awake condition, we measured spine turnover in a group of age-matched animals to the first anesthesia group (S10A and S10C Fig). Moreover, to rule out that the chronic imaging procedure per se and anesthesia in general had a long-lasting effect on the awake-imaging condition, we added another awake-imaging control group with naïve, age-matched animals to the awake-imaging time point in the experimental group (S10A and S10C Fig). In all 3 groups, spine turnover was indistinguishable, indicating that neither age nor previous imaging under anesthesia impacted spine dynamics in the awake-imaging group (S10C Fig).

Next, we asked whether the modulation of spine turnover by GAs was due to acute remodeling of spines during the time of anesthesia. Alternatively, spine turnover might be driven by long-lasting changes in network activity imposed by the slow reversal of all GAs. To capture fast events such as filopodia formation, we acquired image stacks every 10 minutes (Fig 8C). Spine turnover, survival, or density were not significantly altered during the 1 hour of imaging (Fig 8D). Thus, spines were stable during the 1 hour irrespective of the treatment. While mature spines typically show low elimination/formation rates over 1 hour, filopodia are more dynamic [64–66]. Unlike other reports that observed an acute selective formation of filopodia under Keta/Xyl, but not Iso [67], we did not detect any acute effects of GAs on filopodia turnover of CA1 pyramidal cell dendrites. Thus, chronic exposure to all GAs consistently impacted spine dynamics, whereas acute effects were lacking. Keta/Xyl caused a strong decrease in spine turnover, accompanied by a higher surviving fraction and an increased density over time.

## Episodic memory consolidation is impaired by MMF and Keta/Xyl, but not by Iso

Episodic memory formation and consolidation require hippocampal activity. Newly learned experiences are thought of being consolidated via replay events that co-occur with low-

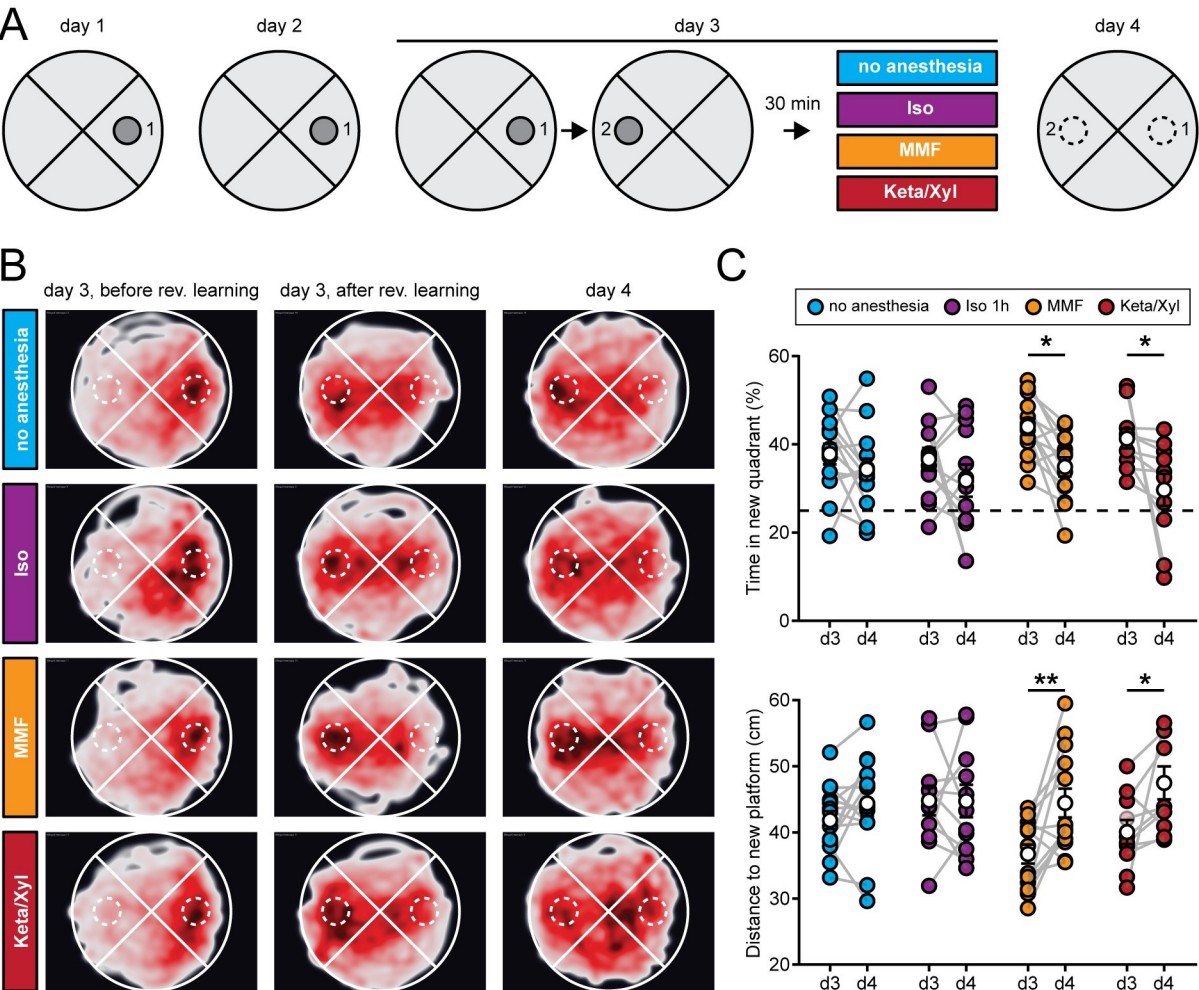

**Fig 9. Episodic memory consolidation is impaired by MMF and Keta/Xyl, but not by Iso. (A)** Experimental design to test episodic-like memory in a Morris water maze. On days 1 and 2, animals were trained to find the platform in position 1. Reversal learning was performed on day 3 where animals had to learn that the platform was moved to position 2. The training was followed 30 minutes later by a 1-hour period of one of the 4 indicated treatments per group. On day 4, consolidation of the memory for the platform in position 2 was tested. **(B)** Heat maps showing trajectories of all mice during the first probe trial before reversal learning on day 3 (left column), after reversal learning on day 3 (middle column), and after treatment on day 4 (right column). The position of the target zone is indicated by dashed circles. **(C)** Scatter plots showing quantification of time spent in the new target quadrant (top) and distance to the new platform (bottom) after reversal learning on day 3 and on day 4. Filled, colored circles indicate individual animals, and white circles indicate mean ± SEM. Asterisks in (C) indicate significant differences between days. * $p < 0.05$, ** $p < 0.01$. For full report of statistics, see S1 Table. All datasets of this figure can be found under https://github.com/mchini/Yang_Chini_et_al/tree/master/Stats_Dataset_(R)/datasets/Figure9_S11. Iso, isoflurane; Keta/Xyl, ketamine/xylazine; MMF, medetomidine/midazolam/fentanyl.

frequency oscillations [49,68–70]. In the hippocampus, these low-frequency events typically occur as sharp waves [50] during sleep, but also during awake resting behavior [70]. The above results from electrophysiological recordings and imaging showed that GAs strongly altered network oscillations in the CA1 area, in the case of MMF and Keta/Xyl, also long after anesthesia discontinuation. Spine turnover of CA1 pyramidal neurons was also affected, especially after Keta/Xyl administration. Therefore, we tested whether inducing anesthesia shortly after the acquisition of a new episodic memory affected its consolidation (Fig 9A). In line with previous experiments, we restricted Iso and MMF anesthesia to 1 hour, while Keta/Xyl anesthesia was left to recede spontaneously. We assessed episodic-like memory with a water maze

protocol for reversal learning, when the hidden platform was moved to the quadrant opposite the initial target location (Fig 9A). Specifically, we tested the effects of the different anesthetics on the consolidation of the memory of the new platform location. We compared the performance of the mice during the probe trial done on day 3 immediately after the reversal learning protocol (and 30 minutes before anesthesia), with the performance during the probe trial on day 4, 24 hours after anesthesia. During the probe trial on day 3, animals of all 4 groups spent significantly more time in the new target quadrant compared to chance (25%), indicating that they learned the new platform position successfully (Fig 9B and 9C).

On day 4, control animals that did not undergo anesthesia showed the same performance as on day 3, suggesting that they had retained the memory of the new platform location (Fig 9B and 9C). However, animals that were anesthetized with Keta/Xyl or MMF spent significantly less time in the new target quadrant and showed a significantly larger mean distance to the target platform position compared to the probe trial on day 3. In the Iso group, no significant difference compared to day 3 was detectable (Fig 9B and 9C, S11A Fig). The impairment of memory consolidation was not explained by the longer duration of recovery after Keta/Xyl or MMF compared to Iso, because anesthesia for up to 4 hours with Iso had no disruptive effect (S11B and S11C Fig). Thus, it is not the duration of the induced unconsciousness but rather the type of anesthetic that likely explains the impaired memory consolidation. Notably, the effects were relatively mild, and the decrease in performance on day 4 was not significantly different between treatment groups. In summary, consistent with long-lasting effects on CA1 network activity, Keta/Xyl and MMF impaired episodic-like memory consolidation. In contrast, Iso, which overall caused a weaker disturbance of neuronal population activity and a faster recovery profile, did not significantly affect memory consolidation.

## Discussion

We investigated and systematically compared the intra- and postanesthetic effects of different commonly used anesthetic strategies on the mouse hippocampus across multiple levels of analysis. Despite sharing some common traits, brain and cellular network states differ substantially under the influence of various types of anesthetics [30,71,72]. Indeed, at the neuronal level, compared with awake state and natural sleep, all 3 anesthetics showed robustly reduced spiking activity in single neurons, reduced power in the high oscillation frequency band, and decorrelated cellular population activity. However, the induced network states in CA1 were highly distinct between the 3 different conditions, with Iso leading to prominent network oscillations at around 0.1 Hz, which timed the spiking activity of single units and neuronal calcium transients. Keta/Xyl caused pronounced oscillations between 0.5 and 4 Hz and the strongest reduction in calcium dynamics. MMF, in contrast, most strongly reduced LFP and SUA and impaired population dynamics for many hours as assessed with calcium imaging. Differences were also present in the long-term effects on spine dynamics, with Keta/Xyl stabilizing spines, leading to reduced turnover and increased density. MMF, on the other hand, mildly increased spine dynamics. Keta/Xyl cannot be antagonized, and therefore, changes of the CA1 network mediated by this anesthetic were present hours after the injection, in agreement with long-lasting overall changes of global animal physiology [38]. More unexpectedly, and in contrast to overall effects on physiology [38], CA1 network dynamics were still disturbed for at least 6 hours after antagonization of MMF anesthesia. These long-lasting alterations were associated with impairment of episodic memory consolidation after exposure to Keta/Xyl or MMF, but not Iso. Thus, despite fulfilling the same hallmarks of general anesthesia, different GAs distinctly alter hippocampal network dynamics, synaptic connectivity, and memory consolidation.

## Iso, MMF, and Keta/Xyl have different molecular targets and distinctly modulate functional and structural features of CA1

The GAs used here represent 3 different strategies based on the large repertoire of currently available anesthetics. Iso represents the class of halogenated diethyl ether analogues, which are volatile and therefore administered via inhalation. Fentanyl, in combination with the analgesic medetomidine and the sedative midazolam (MMF), represents an anesthetic approach based on the injection of a combination of drugs with sedative, analgesic, and anxiolytic properties. In the clinic, propofol can be used instead of midazolam. Finally, ketamine is used both as an anesthetic and, at a lower dosage, as a treatment against depression. For anesthesia, it is generally combined with xylazine, which acts as a sedative, analgesic, and muscle relaxant. All 3 strategies differ markedly in their molecular targets. Consequently, they uniquely modulate general animal physiology [38] and brain activity [72]. Iso is a potent GABA and glycine receptor agonist. Moreover, it activates 2-pore potassium channels and acts as α-amino-3-hydroxy-5-methyl-4-isoxazolepropionic acid receptor (AMPAR) inhibitor [4]. Similar to Iso, midazolam, the hypnotic component of the MMF mix, mainly acts as a GABAR agonist with little effect on NMDARs. In contrast, ketamine is a potent, use-dependent NMDAR blocker with less pronounced effects on potassium channels, GABA, glycine, and other glutamate receptors such as AMPA or kainite receptors [4]. Moreover, while most anesthetics reduce the activity of thalamic nuclei, ketamine increases thalamic drive [73], leading to enhanced rather than reduced oscillations in mid-to-high frequency bands such as theta and gamma [28,74]. In accordance with this, our study reveals major differences in the action of the different anesthetics on functional and structural features of CA1. With both electrical recordings and calcium imaging, we report a robust reduction of neuronal spiking and pairwise neuronal correlation. Notably, effects on electrical activity and calcium activity were well in line for both Iso and MMF, despite the different recording methods. However, we observed some divergence for Keta/Xyl.

## Comparison of electrophysiological recordings and calcium imaging

Generally, differences in electrophysiological recordings and calcium imaging data may stem from the location where the signal is detected. In the calcium imaging experiments, the signal was sampled in a horizontal plane located inside and parallel to stratum pyramidale of CA1. In this configuration, somatic, action potential-driven calcium transients mainly from pyramidal neurons dominate the signal. Due to the kinetics and calcium-binding properties of GCaMP6f, action potentials can only be resolved below approximately 5 Hz and are reported nonlinearly [46,75]. Moreover, expression of the calcium indicator itself influences the detected signals. Low expression levels may bias the detection toward the most active neurons, while strong overexpression of calcium indicators may alter cellular physiology and distort calcium dynamics [76]. In contrast, electrical recordings do not rely on exogenously expressed activity reporters, but require physical insertion of electrodes into the tissue. The electrodes on linear probes are arranged orthogonally to the strata of CA1 and parallel to the dendrites of CA1 cells. Thus, synaptic potentials mainly constitute the LFP across all layers, and spikes are picked up from both pyramidal cells (in stratum pyramidale) and GABAergic neurons in all layers. Moreover, the first method samples neurons that spatially distribute over a large area. In contrast, the second one is biased toward large, active neurons that are in close proximity of the electrode.

More specifically, under Keta/Xyl, the overall firing rate of single units showed the smallest reduction of all 3 anesthetics. At the same time, imaging revealed the most substantial reduction in rate, amplitude, and duration of calcium transients (compare Figs 2B and 3D). One reason for this discrepancy may be the inhibitory action of ketamine on NMDARs. CA1

pyramidal cells display large, NMDAR-driven dendritic plateau potentials and calcium spikes [77]. Moreover, ketamine likely inhibits L-type voltage-gated calcium channels [78] and reduces burst firing [79], leading to calcium transients with reduced amplitude and a faster decay constant. In contrast, ketamine has little influence on sodium spikes and AMPAR-mediated synaptic potentials, which are detected in electrical recordings as SUA and LFP, respectively. In accordance with electrical recordings, calcium transients showed increased power at 0.1 to 0.2 Hz under Iso. However, we did not detect a clear peak at 1 to 4 Hz in the presence of Keta/Xyl, as seen in LFP and SUA, probably due to its strongly dampening effect on calcium transients. The (low-pass) filtering of neuronal activity imposed by calcium indicators might also play a role [75].

Notably, the differences between electrical recordings and calcium imaging under Keta/Xyl are relevant. Calcium is a second messenger central to neuronal plasticity and metabolism [80,81]. NMDARs are a major source for activity-dependent calcium entry into the cell, involved in regulating synaptic plasticity, metabolism, and pathology [82]. The present findings suggest that Keta/Xyl has a particularly strong effect on neuronal calcium activity, uncoupling action potential firing from associated cytosolic calcium transients, leading to reduced intracellular calcium signaling. In contrast, calcium transients under MMF and Iso anesthesia closely matched the electrical activity profile of neurons. Therefore, aside from overall effects on network activity, Keta/Xyl may selectively alter neuronal plasticity by suppressing NMDAR-dependent postsynaptic calcium signals.

## In contrast to neocortex, GAs decorrelate neuronal activity in CA1

All anesthetics decorrelated neuronal activity in CA1, leading to an overall fragmented network state with an increased number of temporal and spatial clusters. This is in contrast with what has been reported from studies on GAs and cortical activity both at adulthood [25–27] and during development [58]. This discrepancy may arise from different ways of analyzing the data. In the present study, all active neurons were identified for each imaging session. Thus, a breakdown of the network results in a larger number of clusters containing less neurons per cluster. In contrast, if a defined population of neurons is followed over time [26], fragmentation of the network may lead to less clusters, since not all neurons remain active during anesthesia. Another reason for the different observations in hippocampus and neocortex may be the distinct architecture of CA1 compared to L2/3 of the neocortex, the latter showing a high degree of local excitatory interconnectivity [47]. In CA1, this is not the case. Pyramidal cells receive their main excitatory input from CA3 and entorhinal cortex and send their efferents to subiculum and extrahippocampal areas without making local connections among each other [9]. Afferent activity originating in various sources and converging in CA1, may arrive out of phase under anesthesia, leading to desynchronized firing of CA1 pyramidal cells. Such a phenomenon has been proposed as a candidate mechanism underlying desynchronization of neuronal firing in basal ganglia under conditions of slow oscillations (slow-wave sleep) and high synchrony in the neocortex [83]. Notably, the pairwise correlation was not entirely independent of the distance between neurons. Synchronization of pyramidal neurons via local, GABAergic interneurons may be another factor that increases spatial correlations. Both in the neocortex and hippocampus, various types of GABAergic interneurons, such as basket or bistratified cells, locally connect to and synchronize pyramidal neurons [84].

Coordinated neuronal network dynamics, including pairwise correlation of calcium transients and single units, population coupling, clustering in the temporal and spatial domain were consistently impaired most strongly with Keta/Xyl and MMF. Iso, both in electrophysiological as well as calcium recordings, showed the mildest effects and permitted hippocampal activity patterns that most closely resembled wakefulness and NREM/REM sleep. Iso and

MMF, in contrast to Keta/Xyl, are thought to be immediately reversible [38]. However, especially MMF showed significant disruption of network dynamics long after reversal both in electrical recordings and with calcium imaging. Antagonization of MMF failed to fully recover calcium dynamics within the following 5 hours. Such long-lasting alterations might interfere with hippocampal function shortly after antagonization of MMF and must be considered when performing whole-cell recordings in freely moving animals [85–87].

Since all anesthetics had a much longer effect on network activity than we expected, we asked whether this is reflected in long-term effects of these different types of anesthetics on spine dynamics of CA1 pyramidal neurons. Recent studies investigating spine dynamics at CA1 pyramidal neurons came to incongruent conclusions reporting spine turnover ranging from 3% [63] over 12% [21] to approximately 80% [62] over 4 days. However, all studies used either Iso- [21] or Keta/Xyl-based [62,63] anesthesia during the repeated imaging sessions. Thus, to what extent anesthesia itself influences spine dynamics is not clear.

## Iso, MMF, and Keta/Xyl distinctly alter spine dynamics in CA1

More generally, various effects of general anesthesia on spine dynamics were reported, depending on the brain region, preparation, age of the animal, and anesthetic strategy. For example, enhanced synaptogenesis has been reported with different types of anesthetics on cortical and hippocampal neurons during development [59,60]. In contrast, one study indicated that spine dynamics were not altered on cortical neurons of adult mice with Keta/Xyl or Iso [67], while another study demonstrated an increase in spine density in somatosensory cortex with ketamine [88]. Also, fentanyl-mediated, concentration-dependent bidirectional modulations of spine dynamics were reported in hippocampal cultures [89].

To systematically compare spine dynamics in CA1 in vivo under different anesthetic treatments, we imaged spines at basal, oblique, and tuft dendrites in a large set of dendrites. We found small, but robust chronic effects of repeated anesthesia. These alterations were present in all strata of CA1, consistent with a layer-independent reduction of SUA during anesthesia. Keta/Xyl decreased spine turnover, leading to a mild increase in spine density over time by stabilizing existing spines. This observation agrees with recent studies that showed a stabilizing effect of ketamine in the somatosensory cortex, resulting in increased spine density [88]. Thus, repeated anesthetic doses of Keta/Xyl may limit overall synaptic plasticity and thus spine turnover. It was further shown that sub-anesthetic, antidepressant doses of ketamine enhance spine density in the prefrontal cortex [90,91], similar to our study of CA1 neurons. Iso and MMF had contrasting effects on spine dynamics compared to Keta/Xyl, mildly enhancing spine turnover, which might be explained by their different pharmacology compared to ketamine, as pointed out above. A second aspect that distinguishes Keta/Xyl from Iso and MMF is its irreversibility, which might lead to longer-lasting alterations of synaptic transmission and E/I ratios leading to differential spine dynamics. This idea is supported by the observation that during the anesthesia period itself, spine turnover was not altered, suggesting that long-lasting and repeated disturbances are required to leave a mark in synaptic connectivity.

## MMF and Keta/Xyl, but not Iso, retrogradely affect episodic-like memory formation

Sleep is a natural form of unconsciousness and is required for memory consolidation, including hippocampus-dependent memories [49,56]. Recent work suggested that sleep- and anesthesia-promoting circuits differ [92,93] while others identified circuit elements shared between sleep and general anesthesia [94], especially during development [58]. Therefore, we asked how the diverse alterations of CA1 network dynamics imposed by the different

anesthetics impact memory consolidation. In our study, Iso resembled most closely network states during wakefulness and natural sleep, while Keta/Xyl and MMF caused strong, lasting alterations of LFP, SUA, and calcium dynamics.

Notably, a single dose of anesthesia with Keta/Xyl and MMF, but not Iso, disrupted memory consolidation using a water maze assay in adult mice. Retrograde amnesia appeared to be more sensitive to the magnitude than the duration of CA1 network disturbance imposed by the various anesthetics. Keta/Xyl and MMF most strongly decorrelated CA1 network activity and reverted only slowly. Extending the duration of Iso anesthesia up to 4 hours, to match the slow recovery after MMF and Keta/Xyl, did not affect memory consolidation. This observation indicates that the slow recovery of network activity after Keta/Xyl and MMF alone cannot explain anesthesia-mediated disruptions of memory consolidation. Instead, specific aspects of the different anesthetics may selectively impact hippocampus-dependent memory formation. For example, ketamine is an NMDAR blocker that has been shown to be necessary for the long-term stabilization of place fields in CA1 [95], encoding of temporal information of episodes [96], and formation of episodic-like memory [97]. Notably, maintenance of Iso anesthesia over much longer time periods eventually also disrupts cognitive performance, suggesting that aside from drug-specific effects, loss of consciousness for extended durations may generally affect synaptic function and memory consolidation [98].

Our results appear at odds with a report [99], where a single, 1-hour treatment with Iso caused deficits in the formation of contextual fear memory, object recognition memory, and performance in the Morris water maze in the following 48 hours. However, this study investigated memory acquisition after anesthesia (i.e., anterograde amnesia), while our study asked whether anesthesia affects the consolidation of a memory formed shortly before the treatment (i.e., retrograde amnesia).

Changes in synaptic connections are considered essential for memory formation and storage [11–14]. Despite a small effect on spine dynamics, the strong and lasting disturbance of hippocampal network activity in CA1 (and most likely other brain areas) by Keta/Xyl and MMF was sufficient to interfere with memory consolidation. The chronic alterations of spine turnover, especially by Keta/Xyl, may therefore indicate that repeated anesthesia can impact long-lasting hippocampus-dependent memories.

To establish a direct link between spine dynamics, network disruptions, and memory, future studies are required that investigate both spine turnover and changes in population coupling at hippocampal neurons causally involved in memory formation and maintenance.

Taken together, we report a novel effect of anesthesia on brain dynamics, namely fragmentation of network activity in hippocampus. We consistently observe this phenomenon across multiple levels of analysis. This unique response compared to the cortex may underlie its high sensitivity to anesthesia, including its central role in amnesia. The extent, duration, and reversibility of network fragmentation depend on the GA used. Therefore, this study may help guide the choice of an appropriate anesthetic strategy, dependent on experimental requirements and constraints, especially in the neurosciences. More generally, our findings might also have relevance for the clinic. Postoperative delirium, a condition that involves memory loss, is still an unresolved mystery. Minimizing the disturbance of hippocampal function may be one building block to overcome this undesired condition.

## Methods

### Experimental models and methods

**Mice.** Adult (3 to 9 months of age) C57BL/6J mice and transgenic Thy1-GFP-M mice of both sexes were used in this study. They were housed and bred in pathogen-free conditions at

the University Medical Center Hamburg-Eppendorf. The light/dark cycle was 12/12 hours, and the humidity and temperature were kept constant (40% relative humidity; 22˚C). Food and water were available ad libitum. All procedures were performed in compliance with German law according and the guidelines of Directive 2010/63/EU. Protocols were approved by the Behörde für Gesundheit und Verbraucherschutz of the City of Hamburg under the license numbers 09/15, 32/17, N18/015, and N19/121.

**Hippocampal recording window surgery and in vivo electrophysiology.** Chronic multi-site extracellular recordings were performed in dorsal CA1 during the dark phase of the dark/light cycle, except for sleep recordings, which were done during the light period. The adapter for head fixation was implanted at least 4 days before recordings. Mice were anesthetized via intraperitoneal injection of MMF and placed on a heating blanket to maintain the body temperature. Eyes were covered with eye ointment (Vidisic, Bausch + Lomb, Berlin, Germany) to prevent drying. Prior to surgery, the depth of anesthesia and analgesia was evaluated with a toe pinch to test the paw withdrawal reflex. Subsequently, mice were fixed in a stereotactic frame, the fur was removed with a fine trimmer, and the skin of the head was disinfected with Betaisodona. After removing the skin, 0.5% bupivacaine/1% lidocaine was locally applied to cutting edges. A metal headpost (Neurotar, Helsinki, Finland) was attached to the skull with dental cement (Super Bond C&B, Sun Medical, Moriyama, Japan), and a craniotomy was performed above the to the dorsal CA1 area (−2.0 mm AP, ± 1.3 mm ML relative to Bregma) which was subsequently protected by a customized synthetic window filled with Kwik-Cast sealant (World Precision Instruments, Friedberg, Germany). After recovery from anesthesia, mice were returned to their home cage and were provided with Meloxicam mixed into soft food for 3 days. After recovery from the surgery, mice were accustomed to head fixation and trained to move in the Mobile HomeCage system (Neurotar). For recordings, craniotomies were reopened by removal of the Kwik-Cast sealant, and multisite electrodes (NeuroNexus, Michigan, United States of America) were inserted into the dorsal CA1 (one-shank, A1x16 recording sites, 50-μm spacing, 1.6-mm deep). A silver wire served as ground and reference in the craniotomy between skull and brain tissue. Extracellular signals were band-pass filtered (0.1 to 8,000 Hz) and digitized (32 kHz) with a multichannel extracellular amplifier (Digital Lynx SX; Neuralynx, Bozeman, Montana, USA). The same animals were recorded weekly under different anesthesia. After 15 minutes of non-anesthetized recording, mice received a subcutaneous injection of Keta/Xyl, MMF, or inhalation of Iso in a pseudo-randomized order. The following drug combinations were administered: 2.0% Iso in 100% $O_2$; 130 mg/kg ketamine, 10 mg/kg xylazine s.c.; 5.0 mg/kg midazolam, 0.2 mg/kg medetomidine and 0.05 mg/kg fentanyl s.c.; and for complete reversal of anesthesia, 0.5 mg/kg flumazenil, 2.5 mg/kg atipamezole, and 0.1 mg/kg buprenorphine s.c. Recordings were conducted for 1.5 hours. After recordings, the craniotomy was closed, and mice were returned to their home cage. Electrode position was confirmed in brain slices postmortem.

**EMG recordings.** EMG electrodes for sleep state classification were implanted during hippocampal recording window surgery. Two gold plates (approximately 3-mm diameter) soldered to epoxy lacquered wires and attached to a connector were inserted into the right and left nuchal muscles and fixed with dental cement (Super Bond C&B, Sun Medical). For EMG recordings, a cable was attached to the implanted connector and directly digitized (32 kHz) and band-pass filtered (8 to 8,000 Hz) through a customized break-out channel board with a multichannel amplifier (Digital Lynx SX; Neuralynx). EMG recordings were done at least for 1 hour and 45 minutes in non-anesthetized mice without disturbances. Mice were recorded 1 to 2 times.

**Virus injection and hippocampal window surgery for in vivo calcium imaging.** C57BL/6J wild-type mice were anesthetized via intraperitoneal injection of MMF and placed on a

heating blanket to maintain the body temperature. Eyes were covered with eye ointment (Vidisic, Bausch + Lomb) to prevent drying. Prior to surgery, the depth of anesthesia and analgesia was evaluated with a toe pinch to test the paw withdrawal reflex. Subsequently, mice were fixed in a stereotactic frame, the fur was removed with a fine trimmer, and the skin of the head was disinfected with Betaisodona. The skin was removed by a midline scalp incision (1 to 3 cm), the skull was cleaned using a bone scraper (Fine Science Tools, Heidelberg, Germany), and a small hole was drilled with a dental drill (Foredom, Bethel, Connecticut, USA) above the injection site. AAV2/7-syn-GCaMP6f was targeted unilaterally to the dorsal CA1 area (−2.0 mm AP, ±1.3 mm ML, −1.5 mm DV relative to Bregma). Moreover, 0.6 µl of virus suspension was injected. All injections were done at 100 nl*min$^{-1}$ using a glass micropipette. After the injection, the pipette stayed in place for at least 5 minutes before it was withdrawn, and the scalp was closed with sutures. For complete reversal of anesthesia, mice received a subcutaneous dose of FAB. During the 2 days following surgery, animals were provided with Meloxicam mixed into soft food. Two weeks after virus injection, mice were anesthetized as described above to implant the hippocampal window. After fur removal, skin above the frontal and parietal bones of the skull was removed by 1 horizontal cut along basis of skull and 2 rostral cuts. The skull was cleaned after removal of the periosteum, roughened with a bone scraper, and covered with a thin layer of cyanoacrylate glue (Pattex, Düsseldorf, Germany). After polymerization a 3-mm circle was marked on the right parietal bone (anteroposterior, −2.2 mm; mediolateral, +1.8 mm relative to bregma) with a biopsy punch, and the bone was removed with a dental drill (Foredom). The dura and somatosensory cortex above the hippocampus were carefully aspirated until the white matter tracts of the corpus callosum became visible. The craniotomy was washed with sterile PBS, and a custom-built imaging window was inserted over the dCA1. The window consisted of a hollow glass cylinder (diameter: 3 mm, wall thickness: 0.1 mm, and height: 1.8 mm) glued to a No. 1 coverslip (diameter: 3 mm and thickness: 0.17 mm) on the bottom and to a stainless-steel rim on the top with UV-curable glass glue (Norland NOA61, Cranbury, New Jersey, USA). The steel rim and a head holder plate (Luigs & Neumann, Ratingen, Germany) were fixed to the skull with cyanoacrylate gel (Pattex). After polymerization, cranial window and head holder plate were covered with dental cement (Super Bond C&B, Sun Medical) to provide strong bonding to the skull bone. Following the surgery, animals were provided with Meloxicam mixed into soft food for 3 days. The position of the hippocampal window was confirmed in brain slices postmortem.

**Two-photon calcium imaging in anesthetized and awake mice.** The same animals were sequentially imaged under Keta/Xyl, MMF, or Iso in a pseudo-randomized order during the dark phase of the dark/light cycle (for details, see above). After losing the righting reflex, generally 5 to 10 minutes after application of the anesthetics, the animals were positioned on a heating pad to maintain body temperature at approximately 37°C during anesthesia. The intensity of anesthesia and evaluation of the different stages of anesthesia were assessed by recording the presence or absence of distinct reflex responses: righting reflex, palpebral reflex, and toe pinch reflex. Between each imaging session, mice were allowed to recover for 1 week.

Anesthetized mice were head-fixed under the microscope on a heated blanket to maintain body temperature. Eyes were covered with eye ointment (Vidisic, Bausch + Lomb) to prevent drying. The window was centered under the 2-photon microscope (MOM scope, Sutter Instrument, Novato, California, USA modified by Rapp Optoelectronics, Wedel, Germany), and GCaMP6f expression was verified in the hippocampus using epi fluorescence. Images were acquired with a 16× water immersion objective (Nikon CFI75 LWD 16X W, 0.80 NA, 3.0 mm WD, Nikon, Amsterdam, the Netherlands). For awake imaging, we used a linear treadmill, which allowed imaging during quiet and running states. Moreover, 5-minute time-lapse images were acquired every 10 minutes for a period of 50 minutes. Only quiet periods were

considered for analysis in this study. Image acquisition was carried out with a Ti:Sa laser (Chameleon Vision-S, Coherent, Dieburg, Germany) tuned to 980 nm to excite GCaMP6f. Single planes (512 × 512 pixels) were acquired at 30 Hz with a resonant galvanometric scanner at 29 to 60 mW (980 nm) using ScanImage 2017b (Vidrio Technologies, Leesburg, Virginia, USA). Emitted photons were detected by a pair of photomultiplier tubes (H7422P-40, Hamamatsu, Herrsching am Ammersee, Germany). A 560 DXCR dichroic mirror and a 525/50 emission filter (Chroma Technology, Bellows Falls, Vermont, USA) was used to detect green fluorescence. Excitation light was blocked by short-pass filters (ET700SP-2P, Chroma Technology). For the repetitive imaging, the position of the FOV was registered in the first imaging session with the help of vascular landmarks and cell bodies of CA1 pyramidal neurons. This allowed for subsequent retrieval of the FOV for each mouse.

Calcium imaging experiments to measure recovery from anesthesia were done in 5 additional animals. They were trained to maintain immobile on the treadmill for extended periods. We ensured to measure the same FOV and to maintain overall stability of fluorescence intensity for every recording in each imaging session for a given animal. The time-lapse recordings were extended to up to a maximum of 10 minutes per time point to have a higher probability of capturing motionless periods continuously, in awake and recovery states. Iso was applied for 60 minutes. FAB was injected 60 minutes after the application of MMF. Keta/Xyl was not antagonized. Imaging of calcium activity was performed before, 0.5, 1.5, 2, 3, 4, 5, and 6 hours after induction of anesthesia. For Iso and MMF, the 1.5-hour time point represented the first imaging session after reversal. Untreated control animals were imaged every hour for the same amount of time.

To habituate mice to sleep under head fixation, we used a linear treadmill, which allowed the mice to move at will. Through the first 4 sessions, mice were kept head-fixed for 15 to 30 minutes. In 10 following sessions, the fixation period was extended up to 4 hours with increasing intervals of 30 minutes. The state of the mouse was continuously monitored with a USB camera, and the running speed was recorded with custom-written scripts in the MATLAB. After habituation to 4-hour head fixation, sleep imaging sessions were recorded, which were synchronized with recordings of the pupil and running speed. Sleep imaging was performed during the light phase of the dark/light cycle.

**Two-photon spine imaging in anesthetized and awake mice.** Three to 4 weeks after window implantation, chronic spine imaging started in Tg(Thy1-EGFP)MJrs/J mice with the first of a total of 4 imaging series (see S10A Fig). Each imaging series was done under one of the 3 anesthetic conditions (Iso, Keta/Xyl, and MMF; see above for details) or during wakefulness. Within 1 series, mice were imaged 5 times every 4 days. Afterwards, mice were allowed to recover for 3 to 4 weeks until the next imaging series under a different anesthetic condition was started. Thus, each experiment lasted approximately 5 months. To avoid time-dependent effects, anesthetic conditions were pseudo-randomized (see S10A Fig). For imaging sessions under anesthesia, mice were head-fixed under the microscope on a heated blanket to maintain body temperature. Eyes were covered with eye ointment (Vidisic, Bausch + Lomb) to prevent drying. The window was centered under the 2-photon microscope (MOM scope, Sutter Instrument, modified by Rapp Optoelectronics), and GFP expression was verified in the hippocampus using epi-fluorescence. Image acquisition was carried out with a Ti:Sa laser (Chameleon Vision-S, Coherent) tuned to 980 nm to excite GFP. Images were acquired with a 40× water immersion objective (Nikon CFI APO NIR 40X W, 0.80 NA, 3.5 mm WD). Single planes (512 × 512 pixels) were acquired at 30 Hz with a resonant scanner at 10 to 60 mW (980 nm) using ScanImage 2017b. Before the first imaging session, we registered the FOVs with the help of vascular landmarks and cell bodies of CA1 pyramidal neurons and selected several regions for longitudinal monitoring across the duration of the time-lapse experiment. Each of these

regions contained between 1 and 2 dendritic segments visibly expressing GFP. The imaging sessions lasted for max 60 minutes, and mice were placed back to their home cages where they woke up.

**Morris water maze.** We designed a protocol for reversal learning in the spatial version of the water maze to assess the possible effects of the different anesthetics on episodic-like memory in mice [100,101]. The water maze consisted of a circular tank (145 cm in diameter) circled by dark curtains and walls. The water was made opaque by the addition of nontoxic white paint such that the white platform (14-cm diameter, 9-cm high, and 1 cm below the water surface) was not visible. Four landmarks (35 × 35 cm) differing in shape and gray gradient were positioned on the wall of the maze. Four white spotlights on the floor around the swimming pool provided homogeneous indirect illumination of 60 lux on the water surface. Mice were first familiarized for 1 day to swim and climb onto a platform (diameter of 10 cm) placed in a small rectangular maze (42.5 × 26.5 cm and 15.5-cm high). During familiarization, the position of the platform was unpredictable since its location was randomized, and training was performed in darkness. After familiarization, mice underwent 3 learning days, during which they had to learn the location of a hidden platform. The starting position and the side of the maze from which mice were taken out of the maze were randomized. On day 1, mice underwent 4 learning trials (maximum duration of 90 seconds, intertrial interval of 10 minutes). After staying on the platform for 15 seconds, mice were returned to their home cage and warmed up under red light. On day 2, mice underwent 2 training trials before they performed a 60-second long probe trial to assess their searching strategy. Afterwards, 1 additional training trial was used to reconsolidate the memory of the platform position, and mice were distributed into 4 groups with a similar distribution of performance. On day 3, the long-term memory of the platform position was tested with a 45-second long probe trial, followed by another training trial with the platform in place to avoid extinction. Then mice underwent 4 reversal learning trials with the platform located in the quadrant opposite the one in which the platform was during the previous training trials. To assess whether the mice learned the new platform position, mice underwent a 60-second long probe trial followed by 1 more training trial to consolidate the memory of the new location. One hour after the last reversal learning trial, mice were anesthetized to analyze the effects of the anesthesia on the consolidation of the memory of the new platform position. Mice were assigned to 4 groups with an equal average performance during the probe trial on day 2. Each group was subjected to different conditions: 1-hour Iso anesthesia, 1-hour MMF anesthesia and Keta/Xyl anesthesia (which was not antagonized), and 1 group was left untreated. On day 4, mice underwent a 60-second long probe trial to evaluate their searching strategies, namely the "episodic-like memory" of the reversal learning trials performed 1 hour before having been anesthetized on day 3 (see Fig 9A).

## Quantification and statistical analysis

**Electrophysiology.** In vivo electrophysiology data were analyzed with custom-written scripts in the MATLAB environment available at https://github.com/mchini/ HanganuOpatzToolbox. We selected the recording site in the pyramidal layer of CA1. Data were band-pass filtered (1 to 100 Hz or 0.1 to 100 Hz for low-frequency LFP analysis) using a third-order Butterworth forward and backward filter to preserve phase information before down-sampling to analyze LFP.

*Detection of active periods.* Active periods were detected with an adapted version of an algorithm for ripple detection (https://github.com/buzsakilab/buzcode/blob/master/detectors/ detectEvents/bz_FindRipples.m). Briefly, active periods were detected on the band-pass filtered (4 to 20 Hz) normalized squared signal using both absolute and relative thresholds. We

first passed the signal through a boxcar filter and then performed hysteresis thresholding: We first detected events whose absolute or relative power exceeded the higher threshold and considered as belonging to the same event all data points that were below the lower (absolute or relative) threshold. Absolute thresholds were set to 7 and 15 μV, relative thresholds to 1 and 2. Periods were merged if having an inter-period interval shorter than 900 ms and discarded if lasted less than 500 ms. Percentage of active periods was calculated for 15-minute bins. Time stamps were preserved for further analysis.

*Power spectral density*. Power spectral density was calculated on 30-second long windows of 0.1 to 100 Hz filtered signal using Welch's method with a signal overlap of 15 seconds.

*Relative change*. Relative change was calculated as (value anesthesia − value pre-anesthetized) / (value anesthesia + value pre-anesthetized).

*Power-law decay exponent of the LFP power spectrum*. The 1/f slope was computed as in [36]. We used robust linear regression (MATLAB function *robustfit.m*) on the log10 of the LFP power spectrum in the 30 to 50 Hz frequency range.

*Phase-amplitude coupling (PAC)*. PAC was calculated on 0.1 to 100 Hz filtered full signal using the PAC toolbox based on relative change measure [102]. Range of phase vector was set to 0.1 to 8 Hz, and range of amplitude vector was set to 20 to 100 Hz. Significant coupling was calculated in comparison to a shuffled dataset. Nonsignificant values were rejected.

*Single unit analysis*. SUA was detected and clustered using klusta [103] and manually curated using phy (https://github.com/cortex-lab).

*Active units*. The recording was divided into 15-minute bins. Single units were considered to be active in the time interval if they fired at least 5 times.

*Pairwise phase consistency*. PPC was computed as previously described [104]. Briefly, the phase in the band of interest was extracted as mentioned above, and the mean of the cosine of the absolute angular distance (dot product) among all single unit pairs of phases was calculated.

*Unit power*. SUA spike trains of each recording were summed in a population vector, and power spectral density was calculated on 30-second long windows using Welch's method with a signal overlap of 15 seconds. The resulting power spectra were normalized by the firing rate in that window.

STTC was computed as previously described [52]. Briefly, we quantified the proportion ($P_A$) of spikes of spike train A that fall within ±Δt of a spike from spike train B. To this value, we subtract the proportion of time that occurs within ±Δt of spikes from spike train B ($T_B$). This is then divided by 1 minus the product of these 2 values. The same is then applied after inverting spike train A and B, and the mean between the 2 values is kept.

$$STTC = \frac{1}{2}\left(\frac{P_A - T_B}{1 - P_A T_B} + \frac{P_B - T_A}{1 - P_B T_A}\right)$$

Importantly, this coefficient has several desirable properties. It is bounded between −1 and 1. It is symmetric with respect to the 2 spike trains. Computing it over different timescales is readily done by controlling the value of the parameter "Δt." Lastly, and most importantly, traditionally used methods of assessing correlations between pairs of spike trains show an inverse correlation between their value and firing rate, due to the fact that spiking is sparse with respect to the sampling frequency, and therefore quiescent period in both spike trains artificially increase the correlation. This is not the case for the STTC [52]. Given that there are large differences in the average firing rate of our conditions, we chose STTC analysis over pure correlation analysis to circumvent this major bias. On the flipside, STTC cannot be straightforwardly applied to negative correlations, which were therefore not investigated in SUA data.

**Calcium imaging data.** In vivo calcium imaging data were analyzed with custom-written scripts in the Python and MATLAB environment available at https://github.com/mchini/Yang_Chini_et_al.

*Alignment of multiple recordings.* To track the activity of the same set of neurons in different anesthetic conditions and during wakefulness, we acquired 2-photon time series of a defined FOV for each animal and each condition across multiple weeks. Over such long time periods, the FOV was susceptible to geometrical transformations from 1 recording to another and thus, any 2 time series were never perfectly aligned. This problem scaled with time that passed between recordings. However, optimal image alignment is critical for the successful identification and calcium analysis of the same neurons across time [105,106].

To address this problem, we developed an approach based on the pystackreg package, a Python implementation of the ImageJ extension TurboReg/StackReg [107]. The source code that reproduces the procedure described in this section is available on github (https://pypi.org/project/pystackreg/). The *pystackreg* package is capable of using different combinations of geometrical transformations for the alignment. We considered rigid body (translation + rotation + scaling) and affine (translation + rotation + scaling + shearing) transformation methods, which we applied to mean and enhanced mean intensity images generated by Suite2p during the registration of each single recording. We performed the alignment using all 4 combinations (2 transformations × 2 types of images) choosing the one with the best performance according to the following procedure. Squared difference between the central part of a reference and aligned image served as a distance function *d* to quantify the alignment (since the signal is not always present on the borders of the image, they were truncated):

$$d = \sum_{i,j}^{\substack{Trunc. \\ images}} \left( x_{i,j}^{ref} - x_{i,j}^{aligned} \right)^2,$$

where $x_{i,j}^{ref}$ and $x_{i,j}^{aligned}$ are intensities of the pixel with coordinates *i,j* of the reference and aligned images. The combination with the smallest score was chosen for the final transformation. In some rare cases, the algorithm of the alignment did not converge for a given transformation method and image type (mean or enhanced mean), crumbling the aligned image in a way that most of the FOV remained empty. This combination may have the smallest distance function *d* and may be falsely identified as the best one. To overcome this issue, an additional criterion was applied, which requires the central part of the aligned picture to contain more than 90% of the non-empty pixels. The overall performance of the algorithm was verified by visual inspection. An example of the alignment of 2 recordings is shown in S5 Fig.

In case of relatively small distortions across recordings, for example, when consecutive acquisitions were done within 1 imaging session, registration can alternatively be performed simultaneously with ROI detection in Suite2p by concatenating those TIFF stacks. In this approach, every ROI is automatically labeled with the same identity (ID) number across all recordings.

*Identification of the same neurons across different recordings and unique neuron ID assignment.* After the alignment procedure, we set out to identify neurons which were active across multiple recordings (and thus, multiple conditions). To achieve this, we developed an algorithm similar to the one described in Sheintuch and colleagues [106].

The algorithm processes in series all recordings for a given animal and assigns unique ID numbers to each ROIs of every recording. Since the recordings under Iso anesthesia had the largest number of active neurons, we chose the first recording of this condition as reference.

We assigned IDs that ranged from 1 to the total amount of neurons to all the ROIs of this recording. For every other recording of each mouse, neuron ID assignment consisted of (1) comparison of the properties (details below) of each ROI with each ROI that had already been processed; (2a) if the properties of the ROI matched the properties of an ROI from a previously analyzed recording, the ROI received the same neuron ID; and (2b) if no match was found, a new (in sequential order) neuron ID was assigned to the ROI. In order to be identified as representing the same neuron in 2 different recordings, 2 ROIs had to respect the following criteria: The distance between their centroids had to be below 3 μm, and the overlap between their pixels had to be above 70%. An example of the identification of unique neuron pairs in 2 recordings is presented in S6A Fig. The thresholds were chosen based on the distribution of the distances between centroids and percentage of the overlaps. An example for a single mouse is graphically illustrated in S6B Fig. Both properties have a clearly bimodal distribution (similar to [106]) with cutoffs close to the chosen thresholds.

*Signal extraction and analysis*. Signal extraction, correlation, and spectral analysis for calcium signal was performed using Python (Python Software Foundation, New Hampshire, USA) in the Spyder (Pierre Raybaut, The Spyder Development Team) development environment. Calcium imaging data were analyzed with the Suite2p toolbox [108] using the parameters given in Table 1.

The same analytical pipeline was applied to both the raw fluorescence traces as well as the deconvolved ("spikes") signal, as extracted by the Suite2p toolbox. Generally, the raw fluorescence signal was preferred over the deconvolved one given that its extraction is more straightforward and relies on less assumptions. However, while the reported effects varied in magnitude depending on which of the 2 signals was considered, the same results were obtained on both datasets. The effects were entirely consistent. For raw signal analysis of each neuron, previous to any further step, we subtracted 0.7 of the corresponding neuropil fluorescence trace.

The number and height of calcium transients properties were calculated with the scipy function *find_peaks* on the raw calcium traces with the following parameters: height = 200, distance = 10, and prominence = 200. The decay was computed on the 10 best isolated transients of each neuron, using the *OASIS* toolbox (https://github.com/j-friedrich/OASIS). We used the *deconvolve* function with the following parameters: penalty = 0 and optimize_g = 10. Traces with an estimated decay over 2 seconds were considered cases of failed extraction and removed from further analysis.

The choice of the parameter values for transient detection is somewhat arbitrary. Similarly, it is debatable whether and how the calcium traces should best be normalized. Therefore, we tested the robustness of our findings by systematically varying signal extraction choices. We first varied the height and prominence threshold across a wide range of values (50 to 700 arbitrary units). We further computed transients features on normalized ΔF/F calcium traces. To normalize calcium signals, we used the baseline value as extracted by the *deconvolve* function. Also, in this case, we varied the height and prominence threshold across a wide range of values (0.5 to 3 arbitrary units). Finally, we computed 2 measures of neuronal activity that are independent of calcium transients detection: the average of the trace integral and its standard deviation, with and without normalization. Across all of these scenarios, the reported effects were robustly consistent.

Correlations were computed both as Pearson (numpy function *corrcoeff*) and Spearman (custom-written function) coefficient on the z-scored signal. To both sets of coefficients, the Fisher correction (the inverse of the hyperbolic tangent function, numpy function *arctanh*) was applied. For power analysis, we first created a population activity vector by summing all the single neuron z-scored signals and then estimated the power spectral density by applying

**Table 1. Parameters used in Suite2p for calcium imaging data analysis.**

| Parameter | Variable | Value |
|---|---|---|
| Sampling rate, frames per second | fs | 30 |
| *Registration* | | |
| Subsampled frames for finding reference image | nimg_init | 2,000 |
| Number of frames per batch | batch_size | 200 |
| Maximum allowed registration shift, as a fraction of frame max (width and height) | maxregshift | 0.1 |
| Precision of subpixel registration (1/subpixel steps) | subpixel | 10 |
| Smoothing | smooth_sigma | 1.15 |
| Bad frames to be excluded | th_badframes | 100.0 |
| *Nonrigid registration* | | |
| Use nonrigid registration | nonrigid | True |
| Block size to register (** keep this a multiple of 2 **) | block_size | [128,128] |
| If any nonrigid block is below this threshold, it gets smoothed until above this threshold. 1.0 results in no smoothing | snr_thresh | 2.0 |
| Maximum pixel shift allowed for nonrigid, relative to rigid | maxregshiftNR | 10 |
| *Cell detection* | | |
| Run ROI extraction | roidetect | True |
| Run sparse_mode | sparse_mode | False |
| Diameter for filtering and extracting | diameter | 12.0 |
| Keep ROIs fully connected (set to 0 for dendrites) | connected | True |
| Maximum number of binned frames for cell detection | nbinned | 5,000 |
| Maximum number of iterations to do cell detection | max_iterations | 20 |
| Adjust the automatically determined threshold by this scalar multiplier | threshold_scaling | 1.0 or 0.1 |
| Cells with more overlap than this get removed during triage, before refinement | max_overlap | 0.75 |
| Running mean subtraction with window of size "high_pass" | high_pass | 100 |
| *ROI extraction* | | |
| Number of pixels to keep between ROI and neuropil donut | inner_neuropil_radius | 2 |
| Minimum number of pixels in the neuropil | min_neuropil_pixels | 100 |
| Pixels that are overlapping are thrown out (False) or added to both ROIs (True) | allow_overlap | True |
| *Deconvolution* | | |
| Deconvolution time constant, seconds | tau | 0.7 |

ROI, region of interest.

the Welch method (sampling frequency = 30 Hz, number of points for fast Fourier transformation = 1,024, and no overlap, window length = 1 second).

For analysis of recovery from anesthesia, all recordings of the imaging session for a given animal were concatenated in Suite2p. As a consequence, each recording in the imaging session has the same set of reconstructed neurons. A time window of 5,000 frames was used for the analysis to ensure continuous motionless periods. To track the neuronal activity changes, the number of fluorescence peaks, their amplitude, and the characteristic decay constant of the transients were considered. Each imaging session's threshold was chosen to match the median activity in the pre-anesthesia (awake) state across all animals. To assess the relative changes of these parameters induced by anesthesia and their subsequent recovery over time, the parameters were normalized to their median value at the pre-anesthesia (awake) state. Notably, we focused our analysis on neurons that maintained some detectable activity during anesthesia, and neurons with no detected peaks were excluded from the distributions. Additionally, we applied the cut *decay constant* > 1/30 [s] (where 30 frames per second is an acquisition rate) to remove the traces where the OASIS algorithm considered a single noise peak to be a calcium transient.

Complexity analysis was performed in the MATLAB (MathWorks) environment. For complexity analysis, we limited the number of neurons to the minimum ($N_{min}$) present in any recording of any condition for each single mouse (median = 265, min = 156, and max = 1068). The resulting matrix therefore had the $T_{rec}xN_{min}$ dimensions, where $T_{rec}$ represents the time vector for the recording, with a length of 5 minutes and a sampling rate of 30 Hz. For recordings that had a number of neurons larger than $N_{min}$ for that mouse, we randomly sampled $n = N_{min}$ neurons and repeated the analysis 5 times. For every extracted parameter, we then considered the median value over the 5 repetitions. For further analysis, the signal was down-sampled from the original sampling frequency of 30 Hz to 10 Hz (100-ms bins). The same analytical pipeline was then applied to both the raw fluorescence traces, as well as the deconvolved signal.

*tSNE clustering.* tSNE clustering was performed similar to [26]. Briefly, in a range between 5 and 45, the perplexity value that minimized the reconstruction error was selected. The number of PCA components used for this step was limited to 30. For the raw fluorescence signal, Euclidian distance was used, whereas for the deconvolved signal, we opted for cosine distance, as it is better suited to a sparse signal. We computed the probability distribution of the resulting embedded matrix ($2xT_{rec}$), that was then convolved with a 2D Gaussian window (standard deviation was set to be equal to 1/40 of the total maximum value). To evaluate the number of clusters in the distribution, we then applied a series of standard steps in image analysis: background subtraction with the rolling ball method, smoothing with a median filter, thresholding, watershedding to avoid undersegmentation, and extended minima transformation. Finally, the exterior boundaries of the objects were automatically traced and counted. This gave the number of clusters.

*Affinity propagation clustering (APC).* APC was performed using a MATLAB toolbox [https://www.psi.toronto.edu/index.php?q=affinity%20propagation] and similarly to [26]. We first obtained a distance map, which was computed as 1 minus the pairwise cosine distance between observations of the $T_{rec}xN_{min}$ matrix. This distance matrix was then fed to the AP algorithm with the input preference set equal to the median of the distance matrix.

*Principal component analysis (PCA) clustering and variance explained.* Standard PCA was applied to the $T_{rec}xN_{min}$ matrix. The number of clusters was computed as the number of components that was needed to cumulatively explain 90% of the variance of the input matrix. Further, we computed the loglog decay coefficient of number of components versus variance explained.

*Community detection.* To detect communities, we used the Louvain algorithm from the Brain Connectivity Toolbox (https://sites.google.com/site/bctnet/), a modularity maximization procedure widely used in studies examining brain networks [109]. This approach aims at subdividing the network into partitions that are more internally dense than would be expected by chance [53]. As input to the algorithm, we used Fisher-transformed correlation matrices obtained from calcium imaging time series. Matrices were not thresholded, and both positive and negative correlations were taken into account to determine optimal modular partitions. The algorithm was evaluated while varying the resolution parameter gamma between 0 and 3, in steps of 0.1. For the multiresolution approach and hierarchical consensus clustering, data were analyzed using code available at https://github.com/LJeub/HierarchicalConsensus and according to the procedure described in [55]. The number of communities detected by the finest level partition of the consensus hierarchy was used for further analysis. While neurons in the awake condition tended to be spatially closer to each other than for the other conditions (S8E Fig), this is unlikely to have influenced the results of the analysis, as the difference was minimal, and there was no correlation between median distance in a recording and the number of detected communities (S8F Fig).

**Sleep scoring.**    Sleep scoring was carried out in 2 steps. We first used electrophysiological features (see below) to classify the behavioral state of the electrophysiological recordings. Then, using this dataset as ground truth, we extracted pupil/eyelid features that we used to extend our classification to the calcium imaging recordings.

*Electrophysiological recordings.* We divided the signal in 30-s epochs with a 50% overlap and used a rule-based approach similar to that applied in [57,110]. NREM sleep epochs were defined as epochs having LFP power in the delta band (1 to 4 Hz) higher than the 70th percentile, EMG broadband (30 to 300 Hz) power lower than the median, and no movement. REM sleep epochs were defined as epochs having a ratio between theta (6 to 12 Hz) and delta LFP power higher than the 70th percentile, EMG broadband power lower than the 25th percentile, and no movement. Finally, wakefulness epochs were defined as epochs having EMG broadband power that exceeded the 80th percentile or with mouse movement. Given that this rule-based approach left approximately 49% of the epochs as unclassified, we extended this classification with a machine learning approach using the scikit-learn toolbox [111]. Using the classified epochs as the labeled dataset, we trained a K-nearest neighbors classifier (number of neighbors = 20, weights = distance, algorithm = auto, leaf size = 5, p = 2, and scoring = f1 macro) to which we fed the following quantile-transformed (quantiles = 20) features: LFP power in the delta and theta band, ratio between LFP delta and theta power, and EMG broadband power and average movement. After training, the algorithm was asked to predict the unclassified epochs. Predictions that were done with a probability estimate above 99% were kept, and the others were left as unclassified. This adjunction to the rule-based approach allowed us to lower the percentage of unclassified (uncertain) epochs to approximately 28%.

*Pupil and eyelid analysis.* During electrophysiological recordings, the eye of the mouse was recorded with a monochrome, infrared sensitive camera (UI-3360CP-NIR-GL Rev. 2, iDS imaging, Germany, objective: LMZ45T3 2/3" 18 to 108 mm/F2.5 manual macro zoom lens, Kowa, Germany) under red light. Videos were captured with the uEye Cockpit software (iDS imaging) with a framerate of 30 Hz. Pupil, EMG, and electrophysiological recordings were synchronized with a customized light/digital pulse shutter. During calcium imaging, eye recordings were done with an infrared sensitive camera (DMK 33UX249; The Imaging Source, Germany) equipped with a macro objective (TMN 1.0/50; The Imaging Source) at a frame rate of 10 Hz. Contours of the mouse eye were tracked using the deep neural network-based software module DeepLabCut [112] and subsequently processed in MATLAB. We trained a neural network (residual neural network, 152 layers, and 200,000 iterations) to detect the upper, lower, left, and right edges of the pupil and eyelid, respectively, in images down-sampled to 256 pixels on the shorter edge ($n$ = 1,038 frames for videos from electrophysiology and $n$ = 2,255 frames for videos from calcium imaging). Besides the position of each tracked point, DeepLabCut provides a value quantifying the certainty about the determined position (which is low in the case of occluded objects, e.g., the pupil during an eyeblink). Samples with a certainty $<$0.5 were linearly interpolated from the last point before to the first point after the respective samples which had a certainty of $>$0.5 (0.24/0.56% of total pupil samples and 0.11/0.12% of total eyelid samples acquired during electrophysiology and calcium imaging experiments, respectively). We then calculated the pupil diameter (as the maximum distance between 2 opposing points of the pupil), as well as its center of mass, and the opening of the eye (as the distance between the top and the bottom eyelid). Finally, blinks were removed from the eye-opening data by linearly interpolating regions which exceed the moving median minus 3 times moving median absolute deviation (sliding window = 30 seconds).

*Calcium imaging recordings.* Using the expanded classification of the electrophysiology dataset, we extracted the following pupil/eyelid features: maximum and minimum pupil diameter, standard deviation of the pupil diameter, pupil area, pupil motion, and eyelid distance.

We then tested the extent to which it was possible to correctly predict the behavioral state on these features alone, similarly to Yüzgec and colleagues. To this aim, we quantile-transformed these features (quantiles = [50, 100, 500]), and passed them to a K-nearest neighbors classifier with similar hyper-parameters as the previous one (number of neighbors = [5, 10], weights = uniform, algorithm = auto, leaf size = 1, p = 2, scoring = f1 macro). Hyper-parameter tuning was done using GridSearchCV. We then iteratively ($n$ = 100) tested the prediction accuracy on 25% of the dataset, yielding good average accuracy for wakefulness (approximately 86%) and NREM sleep (approximately 90%). On the contrary, most REM sleep was mostly classified as NREM (approximately 62%), and the accuracy for this category was significantly lower (approximately 27%). Finally, we retrained the classifier on the entire dataset of pupil/eyelid electrophysiological features and used it to predict the behavioral state of the calcium imaging dataset.

**Two-photon spine image processing.** In each animal, at least 1 GFP-expressing CA1 pyramidal neuron was selected, and 1 to 3 dendrites of 20- to 50-μm length of each of the following types were analyzed: basal dendrites, oblique dendrites emerging from the apical trunk, and tuft dendrites. Motion artifacts were corrected with a custom-modified Lucas–Kanade-based alignment algorithm written in MATLAB. Spines that laterally emanated from the dendrite were counted by manually scrolling through the z-stacks of subsequent imaging time points of the same dendritic element, by an expert examiner blinded to the experimental condition. Protrusions from the dendrite that reached a threshold of 0.2 μm were scored as dendritic spines regardless of shape. If spine neck positions differed 0.5 μm on the subsequent images, the spine was scored as a new spine. Spines were scored as lost if they fell below the threshold of 0.2 μm. Spine density was calculated as the number of spines per μm. The turnover ratio was calculated for every time point by dividing the sum of gained and lost spines by the number of present spines. The survival fraction of spines was calculated as the percentage of remaining spines compared with the first imaging time point.

**Statistical analysis.** Statistical analyses were performed using R Statistical Software (Foundation for Statistical Computing, Vienna, Austria) or GraphPad Prism 9.0 (graphpad.com). All R scripts and datasets are available on GitHub https://github.com/mchini/Yang_Chini_et_al/tree/master/Stats_Dataset_(R)). Nested data were analyzed with linear mixed-effect models to account for the commonly ignored increased false positive rate inherent in nested design [113]. We used "mouse," "recording," "neuron" (calcium imaging), and "single unit" (electrophysiology) as random effects, according to the specific experimental design. Parameter estimation was done using the lmer function implemented in the *lme4* R package [114]. Model selection was performed according to experimental design. Significance and summary tables for lmer model fits were evaluated with the *lmerTest* R package [115], using the Satterthwaite's degrees of freedom method. Post hoc analysis with Tukey multiple comparison correction was carried out using the *emmeans* R package. All statistical analyses can be found at https://github.com/mchini/Yang_Chini_et_al/tree/master/Stats_Dataset_(R)/stats summary.xlsx.

## Supporting information

**S1 Fig. LFP recordings in dorsal CA1 during wakefulness and anesthesia. (A)** Histology images showing the position of the recording electrode in the dorsal hippocampus. Left: Bright-field image merged with fluorescence image of DiI staining. The position of the electrode is indicated by white line drawing. Right: Close-up of dorsal hippocampus showing nuclear DAPI-staining in blue and DiI label from recording electrode position in magenta. **(B)** Heat map displaying raw LFP power in the 0–100 Hz frequency band for the 3 different

anesthetic strategies. **(C)** Average LFP power over time in different frequency bands. Vertical dashed lines in all panels indicate time points of anesthesia induction (Iso, MMF, and Keta/Xyl) and reversal (Iso and MMF only). Lines display mean ± SEM. Asterisks indicate significance of time periods indicated by black horizontal line compared to 15-minute period before anesthesia. Anesthetic conditions are color coded. * $p < 0.05$, ** $p < 0.01$, *** $p < 0.001$. For full report of statistics, see S1 Table. All datasets of this figure can be found under https://github.com/mchini/Yang_Chini_et_al/tree/master/Stats_Dataset_(R)/datasets/Figure1_S1. Iso, isoflurane; Keta/Xyl, ketamine/xylazine; LFP, local field potential; MMF, medetomidine/midazolam/fentanyl.
(TIF)

**S2 Fig. SUA in dorsal CA1 during wakefulness and anesthesia. (A)** Raster plots of z-scored SUA during indicated time periods (black horizontal bars) for the 3 different anesthetic strategies. Units are sorted according to initial activity during wakefulness. Same data as in Fig 2A. **(B)** Scatter plot showing modulation of unit activity during anesthesia with respect to firing rate during wakefulness. *** $p < 0.001$. **(C)** Violin plot showing modulation of unit activity during anesthesia with respect to CA1 anatomical layers. For full report of statistics, see S1 Table. All datasets of this figure can be found under https://github.com/mchini/Yang_Chini_et_al/tree/master/Stats_Dataset_(R)/datasets/Figure2_S2. SUA, single-unit activity.
(TIF)

**S3 Fig. Anesthesia-induced changes in calcium activity profiles are insensitive to the choice of signal extraction parameters. (A)** Histology images showing the position of the imaging window above the dorsal hippocampus. Left: Bright-field image merged with fluorescence. GCaMP6f expression is shown in green. Right: magnified view of the hippocampus. **(B)** Line plot of the number (left) and amplitude (right) of detected calcium transients across varying threshold values used for transient detection. **(C)** Violin plots quantifying the integral (left) and standard deviation (right) of calcium traces. White dots indicate median, and vertical thick and thin lines indicate first to third quartile and interquartile range, respectively. **(D)** and **(E)** As (B, C), for dF/F calcium transients and traces. **(F)** Violin plots quantifying the number (left), amplitude (middle), and decay (right) of calcium transients extracted with the *threshold_scaling* variable set to 0.1 instead of 1. **(G)** tSNE plot summarizing the average calcium transients properties. Each data point represents 1 recording session. Asterisks in (C, E, F) indicate significant differences to wakefulness. *** $p < 0.001$. Note, to facilitate readability, only differences to wakefulness are indicated. For full report of statistics, see S1 Table. All datasets of this figure can be found under https://github.com/mchini/Yang_Chini_et_al/tree/master/Stats_Dataset_(R)/datasets/Figure3_S3. tSNE, t-distributed stochastic neighbor embedding.
(TIF)

**S4 Fig. Oscillations of calcium transients are distinctly altered by Iso, MMF, and Keta/Xyl.** Line plot displaying the spectrograms for population activity power, for raw calcium transients (left) and deconvolved spikes (right) during wakefulness and 3 different anesthetic conditions. All datasets of this figure can be found under https://github.com/mchini/Yang_Chini_et_al/tree/master/Stats_Dataset_(R)/datasets/Figure3_S3. Iso, isoflurane; Keta/Xyl, ketamine/xylazine; MMF, medetomidine/midazolam/fentanyl.
(TIF)

**S5 Fig. Alignment and ROI matching of imaging sessions.** Enhanced mean intensity images are shown to demonstrate the alignment procedure between a recording in Iso **(A)** and one during wakefulness **(B)**. **(C)** Mean intensity image of the awake recording aligned to the

anesthesia condition. **(D)** Overlaid, enhanced mean intensity images before and after **(E)** the alignment algorithm was applied. **(F)** ROIs of active neurons during wakefulness (pink), Iso anesthesia (cyan), and their overlap (black). **(G–I)** Magnified views of a randomly selected sections from panels (D)–(F). Iso, isoflurane; ROI, region of interest.
(TIF)

**S6 Fig. ID assignment and comparison of calcium transients in identified neurons between pairs of conditions. (A)** Time-averaged 2-photon images of the same FOV in CA1 from an "awake" (top left) and an Iso (top right) imaging session aligned to the Iso condition (same images as in Figs 3 and 4). ROIs of active neurons were automatically extracted with a lower quality threshold, accepting more neurons per recording (see Methods). Lower left: geometrical overlay of the 2 images with overlapping ROIs. Lower right: ROIs of identified neurons active in both conditions. **(B)** Kernel density plot and probability density functions for distances between centroids and area overlap for pairs of closest ROIs from the 2 different recordings. A clear bimodal distribution in both parameters is appreciable. Values in the lower right corner indicate highly matched neurons that were considered active in both conditions. **(C)** Kernel density plot and probability density functions for the number (top row), amplitude (middle row), and decay (bottom row) of detected calcium transients for each pair of conditions. From left to right: Iso vs. awake, MMF vs. awake, Keta/Xyl vs. awake, Iso vs. MMF, Iso vs. Keta/Xyl, and MMF vs. Keta/Xyl. Asterisks indicate significant differences between pairs of conditions. Asterisks are always on the side of the unity line with higher values. *** $p < 0.001$. **(D)** Scatter plot showing modulation of calcium transients during anesthesia with respect to calcium activity during wakefulness. *** $p < 0.001$. **(E)** Recovery of CA1 pyramidal neurons from anesthesia. The dots at the beginning of each trajectory represent median values of 2 principal components (PC1 and PC2) calculated on the number, height, and decay of the calcium transients during anesthesia (see Fig 4E). The arrowheads represent the state 6 hours later. Blue dots correspond to awake states in the control group. Every trajectory connects anesthesia and postanesthesia (recovery) states during the following 6 hours for a given animal. The trajectories were obtained using third-degree B-splines. All datasets of this figure can be found under https://github.com/mchini/Yang_Chini_et_al/tree/master/Stats_Dataset_(R)/datasets/Figure4_S6. FOV, field of view; ID, identification; Iso, isoflurane; Keta/Xyl, ketamine/xylazine; MMF, medetomidine/midazolam/fentanyl; ROI, region of interest.
(TIF)

**S7 Fig. Additional correlation analysis. (A)** Violin plots quantifying absolute pairwise correlation of all neurons recorded during calcium imaging shown in Fig 5. **(B)** Decorrelation of calcium activity during anesthesia is sustained also in neurons active during all conditions. Left: Line plot displaying cumulative distribution of Fisher-corrected Pearson correlation coefficients between the same pairs of neurons, which were active in each condition. Right: Violin plots quantifying absolute pairwise correlation coefficients. In violin plots, white dots indicate median, and vertical thick and thin lines indicate first to third quartile and interquartile range, respectively. **(C)** Quantification of correlation between pairs of extracellularly recorded single units using the STTC with shorter integration window. Left: Schematic illustration the quantification of the STTC. Middle: cumulative distribution of the STTC with a 10-ms integration window. Right: violin plot quantifying the tiling coefficient. In all violin plots, white dots indicate median, and vertical thick and thin lines indicate first to third quartile and interquartile range, respectively. Asterisks indicate significant differences to wakefulness. * $p < 0.05$, *** $p < 0.001$. Note, only differences to wakefulness are indicated. For comparison between conditions, see S1 Table. All datasets of this figure can be found under https://github.com/mchini/Yang_Chini_et_al/tree/master/Stats_Dataset_(R)/datasets/Figure5_S7. STTC, spike time tiling

coefficient.
(TIF)

**S8 Fig. Clustering analysis of deconvolved calcium imaging data. (A)** Violin plot quantifying the number of tSNE clusters obtained from deconvolved calcium recordings (i.e., "spikes") during wakefulness, Iso, MMF, and Keta/Xyl anesthesia. **(B)** Violin plot quantifying the number of clusters obtained by AP from deconvolved calcium recordings (i.e., "spikes") during the 4 different conditions. **(C)** Violin plot quantifying the number of communities obtained by MRCC for the 4 different conditions. **(D)** Line plot quantifying maximum community cluster size normalized by the total number of neurons across the resolution parameter gamma ranging from 0 to 3. **(E)** Cumulative distribution of the distance between neurons randomly selected for the spatial cluster analysis showing no difference between conditions. **(F)** Scatter plot displaying absence of a relationship between the median distance of neurons and the number of detected communities. Horizontal lines in violin plots indicate median and first to third quartile. Asterisks in (A)–(C) indicate significant differences to wakefulness. $^{*} p < 0.05$, $^{**} p < 0.01$, $^{***} p < 0.001$. Horizontal lines above plots in (D) and (E) indicate significant difference to wakefulness. Anesthetic conditions are color coded. Note, only differences to wakefulness are indicated. For comparison between conditions, see S1 Table. All datasets of this figure can be found under https://github.com/mchini/Yang_Chini_et_al/tree/master/Stats_Dataset_(R)/datasets/Figure6_S8. APC, affinity propagation clustering; Iso, isoflurane; Keta/Xyl, ketamine/xylazine; MMF, medetomidine/midazolam/fentanyl; MRCC, multiresolution consensus clustering; tSNE, t-distributed stochastic neighbor embedding.
(TIF)

**S9 Fig. Sleep scoring.** Sleep scoring was carried out in 2 steps. We first used electrophysiological features to classify the behavioral state of the electrophysiological recordings. Then, using this dataset as ground truth, we extracted pupil/eyelid features that we used to expand our classification to the calcium imaging recordings. **(A)** Wake, NREM and REM sleep epochs were classified based on animal motion, neck muscle EMG broad power, MUA, and several LFP features. **(B)** Using the electrophysiology-based classification, the following pupil/eyelid features were extracted to classify sleep: maximum and minimum pupil diameter, standard deviation of pupil diameter, pupil area, pupil motion, and eyelid distance. **(C)** Confusion matrix and prediction accuracy for the classification of sleep periods based on eye imaging alone. **(D)** Probability distribution of pupil diameter for predicted wake, NREM, and REM periods in the calcium imaging dataset. **(E)** Classification of activity states in individual mice during electrical recordings (left) and calcium imaging (right). Note that different animals were used for electrophysiology and calcium imaging. All datasets of this figure can be found under https://github.com/mchini/Yang_Chini_et_al/tree/master/Stats_Dataset_(R)/datasets/Figure7_S9. EMG, electromyography; LFP, local field potential; MUA, multi-unit activity; NREM, non-rapid eye movement; REM, rapid eye movement.
(TIF)

**S10 Fig. Chronic spine imaging. (A)** Experimental scheme for chronic spine turnover measurements. Spine imaging was performed in a pseudo-randomized order for the different anesthetics followed by imaging during wakefulness. Each colored box indicates 1 imaging session. For each condition, imaging was done 5 times every 4 days, followed by a 1-month break. To control for long-term effects of anesthesia and age on the awake condition, we performed imaging only during wakefulness in additional mice as indicated. **(B)** Dot plots showing quantification of overall gain and loss of spines during chronic imaging under the 4 different treatments. Dots indicate mean ± SEM. **(C)** Dot plots showing quantification of spine turnover

(left), spine survival (middle), and spine density (right) during wakefulness after anesthetic treatments (same data as "awake" in Fig 7B) and wakeful imaging in the 2 control groups ("awake early" and "awake late"). Dots indicate mean ± SEM. **(D)** Dot plots showing quantification of spine turnover (left column), spine survival (middle column), and spine density (right column) separately for basal (top row), oblique (middle row), and tuft dendrites (bottom row). Same data as in Fig 8B. Asterisks indicate significant differences to wakefulness. $^*$ $p < 0.05$, $^{**}$ $p < 0.01$. For full report of statistics, see S1 Table. All datasets of this figure can be found under https://github.com/mchini/Yang_Chini_et_al/tree/master/Stats_Dataset_(R)/datasets/Figure8_S10.
(TIF)

**S11 Fig. Comparison of the effect of Iso, MMF, and Keta/Xyl on episodic memory consolidation. (A)** Scatter plots showing quantification of change in the time spent in the new target quadrant (left) and distance to the new platform (right) on day 4 after 1 hour of indicated anesthesia or no treatment compared to day 3. **(B)** Scatter plots showing quantification of time spent in the new target quadrant (left) and distance to the new platform (right) after reversal learning on day 3 and on day 4 after 1, 2, and 4 hours of Iso anesthesia. Filled, colored circles indicate individual animals. White circles indicate mean ± SEM. **(C)** Scatter plots showing quantification of change in the time spent in the new target quadrant (left) and distance to the new platform (right) on day 4 after 1, 2, and 4 hours of Iso anesthesia compared to day 3. Asterisks indicate significant deviation from 0. $^*$ $p < 0.05$, $^{**}$ $p < 0.01$. Note, significant differences between groups were not evident. For full report of statistics, see S1 Table. All datasets of this figure can be found under https://github.com/mchini/Yang_Chini_et_al/tree/master/Stats_Dataset_(R)/datasets/Figure9_S11. Iso, isoflurane; Keta/Xyl, ketamine/xylazine; MMF, medetomidine/midazolam/fentanyl.
(TIF)

**S1 Table. Statistics summary.**
(XLSX)

## Acknowledgments

We thank Stefan Schillemeit and Kathrin Sauter for technical assistance and Thomas G. Oertner and Amit Marmelshtein for critical feedback on the manuscript.

## Author Contributions

**Conceptualization:** Wei Yang, Mattia Chini, Jastyn A. Pöpplau, Fabio Morellini, Olaf Sporns, Ileana L. Hanganu-Opatz, J. Simon Wiegert.

**Data curation:** Wei Yang, Mattia Chini, Jastyn A. Pöpplau, Andrey Formozov, Alexander Dieter.

**Formal analysis:** Wei Yang, Mattia Chini, Jastyn A. Pöpplau, Andrey Formozov, Alexander Dieter, Patrick Piechocinski, Cynthia Rais, Fabio Morellini, Olaf Sporns, J. Simon Wiegert.

**Funding acquisition:** Wei Yang, Andrey Formozov, Ileana L. Hanganu-Opatz, J. Simon Wiegert.

**Investigation:** Wei Yang, Jastyn A. Pöpplau, J. Simon Wiegert.

**Methodology:** Wei Yang, Mattia Chini, Jastyn A. Pöpplau, Andrey Formozov, Alexander Dieter, Fabio Morellini, Olaf Sporns, J. Simon Wiegert.

**Project administration:** J. Simon Wiegert.

**Resources:** Ileana L. Hanganu-Opatz, J. Simon Wiegert.

**Software:** Mattia Chini, Andrey Formozov, Alexander Dieter, Olaf Sporns.

**Supervision:** Ileana L. Hanganu-Opatz, J. Simon Wiegert.

**Writing – original draft:** Wei Yang, Mattia Chini, J. Simon Wiegert.

**Writing – review & editing:** Wei Yang, Mattia Chini, Ileana L. Hanganu-Opatz, J. Simon Wiegert.

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
