## [Editor Report · Decision Letter 0]

8 Jul 2020

Dear Dr Wiegert, 

Thank you for submitting your manuscript entitled "Anesthetics uniquely decorrelate hippocampal network activity, alter spine dynamics and affect memory consolidation" for consideration as a Research Article by PLOS Biology.

Your manuscript has now been evaluated by the PLOS Biology editorial staff, as well as by an academic editor with relevant expertise, and I'm writing to let you know that we would like to send your submission out for external peer review.

Please re-submit your manuscript within two working days, i.e. by Jul 10 2020 11:59PM.

Kind regards,

Roli Roberts PhD

Senior Editor

PLOS Biology

on behalf of

Gabriel Gasque, Ph.D.,

Senior Editor

PLOS Biology

---

## [Decision Letter · Decision Letter 1]

15 Sep 2020

Dear Dr Wiegert,

Thank you very much for submitting your manuscript "Anesthetics uniquely decorrelate hippocampal network activity, alter spine dynamics and affect memory consolidation" for consideration as a Research Article at PLOS Biology. Your manuscript has been evaluated by the PLOS Biology editors, an Academic Editor with relevant expertise, and by three independent reviewers. Please accept my apologies for the unusual time that peer review is taking during the difficult summer months and the pandemic.

You'll see that while the reviewers are somewhat positive, reviewers #1 and #2 raise some significant concerns that you will need to address for further consideration. In case you find their comments useful, the Academic Editor said, "I think both reviewer #1 and #2's comments are important. Perhaps a longer anesthesia for the volatile anesthetics could be a solution to check whether the behavioral effects still remain. And I completely agree that it is more meaningful to compare sleep LFP/firing rates with anesthesia and it can be done - of course i would not expect them to do the imaging during sleep. I think we can let the authors to revise if they are willing to perform the extra experimental work. Whether regular paper or resource, i think i depends whether they can strengthen the behavioral findings in the light of reviewer #1's comment."

In light of the reviews (below), we will not be able to accept the current version of the manuscript, but we would welcome re-submission of a much-revised version that takes into account the reviewers' comments. We cannot make any decision about publication until we have seen the revised manuscript and your response to the reviewers' comments. Your revised manuscript is also likely to be sent for further evaluation by the reviewers.

We expect to receive your revised manuscript within 3 months. 

**IMPORTANT - SUBMITTING YOUR REVISION**

*Re-submission Checklist*

*Published Peer Review*

*PLOS Data Policy*

*Blot and Gel Data Policy*

Sincerely,

Roli Roberts

Senior Editor,

rroberts@plos.org,

PLOS Biology

REVIEWERS' COMMENTS:

Reviewer #1:

This is a very detailed analysis of the effect of three general anesthetics on hippocmapal functions (isoflurane, medetomidine/midazolam/fentanyl and ketamine/xylazine). Authors use multisite recordings of LFP and units, multicellular and spine calcium imaging and behavioral tasks to examine their effects longitudinally in head-fixed mice (except for behav tasks). They found different effects from mild (iso) to disturning (keta/x) on indices of neuronal firing (LFP and unit), correlational firing (calcium imaging, units), spine turnoever (calcium imaging) and episodic memory consolidation (watermaze). They conclude different anesthetics affect hippocampal network dynamics and function differently. 

While the issue is not novel, the study is quite compelling in terms of the different preparations and the longitudinal design used to test the effect of the same anesthetics in all animals. The topic is of interest given prevalence of post-operative cognitive decline. I have however several comments regarding the analysis and concerns regarding the reach of conclusions. 

Major:

1- The main conclusion that keta/x works worse than iso and MMF may be quite naive. Iso is a volatile anesthetics and therefore it is not surprising recovery is quite straitforward. MMF can be quickly antagonized as well. This different recovery trends of course have different impact on post-anesthetics cognitive performance. Also, the effect is not only neural, but systemic, affecting the entire organism. So, it is a bit unclear how the different pieces of the analysis can related one each other in a more 'causative' way (ie. whether epidosic memory deffects can be explained by changes in LFP, units, spines...). This is my major concern regarding the reach of conclusions. Actually, in their behav experiments authors directly declare there are differences between groups in terms of the recovery time (page 17, line 539). To my eyes that preclude any possibility to compare cognitive effects and their conclusions on this block are missleading.

2-More specifically, I have some additional comments and concerns that require clarification. First, where are LFP recorded in the dorsal hippocampus? There are laminar variabilities along the structure and I guess authors have homogenized their analysis to specific strata (i.e. SLM?). If so, this should be clarified. Importantly, by collecting LFP signal from only one strata you may miss effects ocurring at other layers, especially if the action of the anethetics differs in terms of excitatory or inhibitiory pathways, for instance. Later in their analysis of dendritic spines authors aim to evaluate different layers but the reader is left with the impression that laminar differences were not appropiately controlled. I recommend authors more consistently including results per lamina in their analysis (i.e. SO, SR, SLM). Here too, having access to spatial LFP signals allow to estimate current-source densities (CSD) which provide more specific information regarding the main input pathways than LFP

3-I am unclear how authors have collected information from slow activities given their amplifiers are band-pass filtering signals and operating in AC-coupled. Most of the power spectrum <0.1Hz is confusing to me and does not feel real (directly measured), but the result of some extrapolation. This should be clarified and addressed.

4- Using the 1/f spectral trend to evaluate E/I ratio is not standard. Authors stand on the Gao et al. 2017 paper but this measurement is not widely accepted. Actually, the 1/f trend more likely reflect the properties of the tissue-electrode interface with the amplifier dominating responses (i.e. one part of 1/f is the electric noise). Biological deviation from that noise reflect physiolopgical activity, and so the measure is circular. I recommend authors using other types of estimation (like interneuron/pyramidal cell rate, etc...)

5- PPC (Vinck et al.) analysis on Fig.2. depends on the firing rate. Because there are differences in the FR between groups I recommend authors reinforcing this part with an additional measure (i.e. their STTC used later in the ms). 

6- For the cluster analysis and results it is unclear whether effects depend on one animal or are consistent across mice (Fig.6). A more consistent reporting per animal should be added as suplementary information here.

7- The effects described here may also depend on the age of the animal, gender and invidivual variability of cognitive trends prior to anesthesia (although those can be homogenized). While this is commented in the ms, the reader is left with the impression that there are other critical axes than the one examined here.

Reviewer #2:

This manuscript describes the effects of general anesthetics on hippocampal activity, spine turnover, and episodic memory. The authors found that isoflurane (Iso), medetomidine/midazolam/fentanyl (MMF), and ketamine/xylazine (Keta/Xyl) distinctly affected network activity measured by local field potentials, spiking activity, calcium events, spike dynamics in the hippocampal CA1. In addition, they showed that the anesthetics distinctly induced retrograde memory impairments measured in the Morris water maze. I think the analysis were well done, and this manuscript would be a useful catalogue of effects of general anesthetics for the future animal research and clinical settings. However, when thinking about the relationship between hippocampal activity, spine turnover, and episodic memory, the manuscript does not make a strong link between them or shed new light on a major problem in the field, and this I feel is a main shortcoming of the manuscript for PLoS Biology. 

To enhance the manuscript, I would like to suggest the followings:

1) The manuscript compared the hippocampal activity during waking conscious state and unconscious states under general anesthetics control, and all the anesthetics examined showed distinct activity compared with awake. This is fine, but I believe that more important and intriguing comparison would be between hippocampal activity during natural sleep and under anesthesia, since both periods are unconscious states but only the latter is associated with transient amnesia. In addition, is the neuronal activity under the control of general anesthetics more similar to that in non-REM sleep, which is unconscious state, than that in REM sleep, which is conscious state?

2) Firing rates of individual hippocampal pyramidal neurons span at least 3 orders, and neuronal activities of high- and low- firing rate neurons are distinctly affected by natural sleep (Miyawaki and Diba, Current Biology, 2019). Comparison between effects of general anesthetics on high- and low firing rate neurons would give us hints regarding similarity and difference between natural sleep and under anesthesia.

Reviewer #3:

[see also attached formatted version]

The authors of the current study investigated and compared the effects of three general anesthetics, isoflurane (Iso), medetomidine/midazolam/fentanyl (MMF), and ketamine/xylazine (Keta/Xyl), on neuronal activity of the mouse hippocampus and memory consolidation at the behavioral level. The neuronal activity was investigated at the local field potential, single unit electrophysiology level, and using chronic two-photon microscopy to monitor calcium transients in large ensembles of neurons and related structural dendritic spine dynamics. The main conclusion is that all three anesthetics reduced neuronal firing rates, decorrelated pairwise neuronal activity, and altered spine dynamics though with distinct signatures. In particular, Iso anesthesia resembled most closely the neuronal activity during wakefulness, was followed by the fastest network recovery, and, contrary to the other two anesthetics, did not impair consolidation of the memory encoded prior to its administration. The authors conclude that the three anesthetics are distinguished in their main effects on hippocampal activity, with implications for their future selection as anesthetics of election for animal research and clinical settings. Overall, the experiments are well designed, the data support the conclusions, the data analysis is comprehensive and appropriate, and the findings are novel (and somewhat in contrast with previous results from the neocortex) and interesting for the broad readership of the journal. In addition, the results have wide implications, particularly when considering that invasive animal research cannot avoid the use of general anesthetics, most of which are comprehensively examined in this study. I only have minor comments that are aimed at improving the overall impact of the study.

 1. There are a number of statements for which citations are given, but facts are omitted. The authors should carefully proofread the manuscript and made the facts directly available to the reader (rather than asking the reader to consult the citation). I only list a couple of examples: 

 - line 71: "…relaying information about external and internal states, respectively". It is not clear what the authors refer to as internal and external nor whether the CA3 and EC areas convey that type of information to CA1 and how. The authors should elaborate these ideas in the Introduction

 - lines 141-144: the authors should clarify whether the wake-up cocktail refers only to MMF anesthesia or they think it may also impact Iso anesthesia

 - line 70: it would help to mention stratum lacunosum-moleculare as the target for EC3 input to CA1

2. The same animals were administered with all three anesthetics. It is not clear, nor discussed, the implications of previous anesthesia and of its type on the following ones when it comes to hippocampal network activity. The authors should try to compare the network effects of first-time anesthesia with those of the second-time and more generally with the history of multiple anesthetics used. This would likely add relevance and rigor to the study.

3. There is a body of studies that used wake-up cocktails to study neuronal activity (e.g., intracellular recording of CA1 pyramidal cells in freely-moving rats) and other that used NMDA blockers at subanesthetic doses to study stability or replay of place cell activity in the hippocampus. The authors should connect the results of their current study to some of the findings reported by the other aforementioned studies and comment on the potential implications of using the mentioned protocols on hippocampal activity.

4. The violin plots are used extensively throughout the manuscript. They are good at illustrating data with large ranges, but sometimes are difficult to read. Whenever possible, I would suggest converting those plots to more standard bar graphs with overlaying dot-plots or box-and-whiskers.

5. Some of the two-photon images (Fig. 3B and 4A) appear rotated and re-aligned compared to their counterparts. The authors should explain if the field movement was restricted to the 2-D field or additional 3-D alignment was needed and also why the images under MMF anesthesia appear so different compared with the other conditions.

---

## [Decision Letter · Decision Letter 2]

5 Feb 2021

Dear Dr Wiegert,

Thank you for submitting your revised Research Article entitled "Anesthetics fragment hippocampal network activity, alter spine dynamics and affect memory consolidation" for publication in PLOS Biology. I have now obtained advice from two of the original reviewers and have discussed their comments with the Academic Editor. 

Based on the reviews, we will probably accept this manuscript for publication, assuming that you will modify the manuscript to address the remaining points raised by the reviewers. Please also make sure to address the data and other policy-related requests noted at the end of this email.

IMPORTANT:

a) Please attend to the remaining requests from reviewer #1.

b) Please attend to my Ethics and Data Policy requests further down.

We expect to receive your revised manuscript within two weeks. Your revisions should address the specific points made by each reviewer. 

-  a cover letter that should detail your responses to any editorial requests, if applicable

*Published Peer Review History*

*Early Version*

Sincerely,

Roli Roberts

Senior Editor,

rroberts@plos.org,

PLOS Biology

ETHICS STATEMENT:

We note that your Ethics Statement currently says " “All procedures were performed in compliance with German law according and the guidelines of Directive 2010/63/EU. Protocols were approved by the Behörde für Gesundheit und Verbraucherschutz of the City of Hamburg.” Please could you also include an approval number.

DATA POLICY:

Many thanks for depositing your raw data in GIN, and your code in GitHub; this is much appreciated. However, we also ask that all individual quantitative observations that underlie the data summarized in the figures and results of your paper be made available, either as Supplementary Files or by deposition in a publicly available repository.

I note that some of your Fig panels may be plotted directly from the GIN deposition, and I also note that you provide a supplementary file called “stats summary revision.xlsx,” but it's unclear how much this represents the actual numerical values displayed or merely the stats analysis. Please could you clarify this, and provide underlying data if these are not in either of these sources. NOTE: the numerical data provided should include all replicates AND the way in which the plotted mean and errors were derived (it should not present only the mean/average values).

REVIEWERS' COMMENTS:

Reviewer #1:

Authors have addressed my comments with new data and analysis. I found the current version is improved. 

I have only few more comments regading data analysis and presentation:

- Violin plots depend on a kernel density function. For cases of somehow homogenegous distribution there is no big issue, but this is not the case for uneven distributions such as in Fig.6. Can authors specify the statistical design shown here? Is the number of animals nested? Is there one particular animal influencing more results than another? Is grouping showing this nested variability? Consider reporting mean data per mouse.

- Same for Fig.7H,K,L. Can these data be log-normalized? Here too, regarding sleep classification (Fig.7A,I), please provide data point distribution between mice.

Reviewer #2:

I believe that all the issues pointed out by the reviewers have been adequately addressed and resolved.

---

## [Editor Report · Decision Letter 3]

15 Feb 2021

Dear Dr Wiegert,

On behalf of my colleagues and the Academic Editor, Jozsef Csicsvari, I'm pleased to say that we can in principle offer to publish your Research Article "Anesthetics fragment hippocampal network activity, alter spine dynamics and affect memory consolidation" in PLOS Biology, provided you address any remaining formatting and reporting issues. These will be detailed in an email that will follow this letter and that you will usually receive within 2-3 business days, during which time no action is required from you. Please note that we will not be able to formally accept your manuscript and schedule it for publication until you have made the required changes.

PRESS: We frequently collaborate with press offices. If your institution or institutions have a press office, please notify them about your upcoming paper at this point, to enable them to help maximise its impact. If the press office is planning to promote your findings, we would be grateful if they could coordinate with biologypress@plos.org. If you have not yet opted out of the early version process, we ask that you notify us immediately of any press plans so that we may do so on your behalf.

Thank you again for supporting Open Access publishing. We look forward to publishing your paper in PLOS Biology. 

Sincerely, 

Roli Roberts

Roland G Roberts, PhD 

Senior Editor 

PLOS Biology